# TIME-SERIES CAUSAL DISCOVERY VIA DIFFERENTIABLE PERMUTATIONS

## ABSTRACT

Causal discovery with instantaneous effects in multivariate time series is challenging, as the instantaneous structure must be acyclic. Prior methods enforce this by either separating instantaneous and lagged estimation into multi-stage pipelines or imposing algebraic acyclicity constraints via complex augmented Lagrangian optimization, both of which incur high computational cost. In this work, we propose a different approach: we learn a differentiable permutation of variables using the Gumbel–Sinkhorn operator and triangularize the instantaneous coefficient matrix of a Structural Vector Autoregressive (SVAR) model in the learned order. This converts acyclicity from a hard constraint into a parameterization and keeps it valid throughout optimization. In doing so, our method enables unified, continuous optimization with gradient-based learning, leading to improved efficiency in time series causal discovery. Across three real-world benchmarks, our method achieves the best overall performance compared with 12 baselines in both discovery accuracy and efficiency. On the large-scale benchmark, it further demonstrates strong scalability, achieving more than a 6× speedup over competing methods.

## 1 INTRODUCTION

Time-series causal discovery helps recover cause–effect relationships in dynamical systems, and is widely applied in diverse fields such as economics (Hoover, 2001), earth science (Runge et al., 2019a), and industrial systems (Mogensen et al., 2024). As noted by Assaad et al. (2022b), true causal discovery methods for time series should account for both instantaneous causal effects, where $x_{i,t}$ affects $x_{j,t}$ within the same step; and lagged causal effects, where a past state $x_{i,t-\tau}$ (with $\tau > 0$) influences a future state $x_{j,t}$. These relationships are typically formalized as directed graphs. The main goal of causal discovery is to build a causal graph from observed data (Assaad et al., 2022b).

To ensure validity, the causal graph needs to satisfy acyclic constraints (as shown in Figure 1). In particular, for lagged effects, one could easily avoid cycles by avoiding causal links from the future to the past. For instantaneous effects, however, cycles may occur if two variables affect each other within the same time step. To avoid this, instantaneous effects are typically constrained to form a Directed Acyclic Graph (DAG) to guarantee identifiability of the causal structure (Kilian, 2006).

The central challenge in time-series causal discovery is therefore enforcing acyclicity on instantaneous effects. Among existing methods that explicitly enforce acyclicity, two main strategies exist: (1) discrete combinatorial search for a causal order, followed by estimation of causal effects based on the obtained order, as in VARLiNGAM (Hyvärinen et al., 2010) and TiMINo (Peters et al., 2013); and (2) algebraic acyclicity constraints enforced by an augmented Lagrangian optimization while estimating causal effects, as in DYNOTEARS (Pamfil et al., 2020).

While effective, these approaches have notable limitations: (1) they rely on hard acyclicity constraints: order-based methods fix a causal order upfront that cannot adapt even if it poorly fits the data, and augmented Lagrangian methods enforce strict algebraic constraints that require optimization until near-exact satisfaction. (2) This rigidity forces a multi-stage process, separating acyclicity enforcement from causal-effect estimation and thereby risking error propagation, as discussed in (Pamfil et al., 2020). (3) They also incur high computational cost: augmented Lagrangian optimization involves nested loops with unpredictable iterations, and multi-stage pipelines are less amenable to scalable gradient-based learning. These limitations motivate the need for more flexible, unified methods that efficiently enforce instantaneous acyclicity and scale to high-dimensional data.

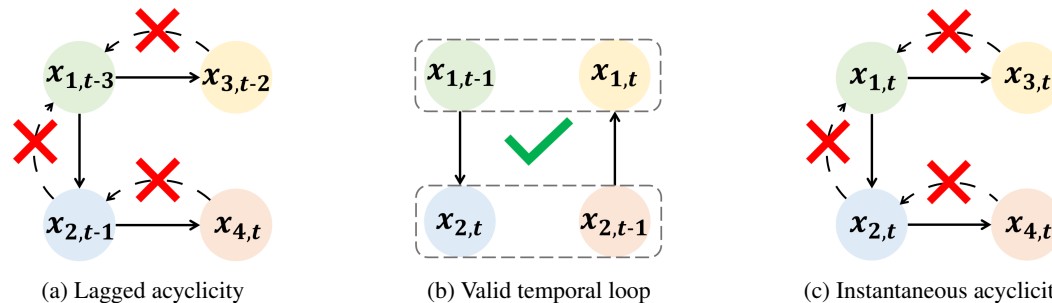

(a) Lagged acyclicity      (b) Valid temporal loop      (c) Instantaneous acyclicity

Figure 1: Illustration of acyclicity in time-series causal graphs, where nodes denote variables and arrows indicate directional influences between variables. (a) Lagged acyclicity: cycles cannot occur across lagged dependencies because future states cannot cause the past. (b) Valid temporal loop: feedback loops across time are allowed as long as all arrows follow temporal order. (c) Instantaneous acyclicity: instantaneous cycles within the same time step are forbidden to ensure identifiability.

**Contributions** We propose a new time-series causal discovery method based on a Structural Vector Autoregressive (SVAR) model (Swanson & Granger, 1997; Demiralp & Hoover, 2003; Moneta & Spirtes, 2006), where causal structure learning is reformulated as the problem of fitting SVAR coefficients, from which the causal graph can be directly constructed. Our main contributions include:

- **Soft acyclicity** (see Section 4.3). We cast acyclicity as a permutation learning problem and design a Gumbel–Sinkhorn operator to relax the permutation into a differentiable form. This enables the causal order to be learned adaptively during optimization rather than fixed upfront, making it soft, data-driven, and dynamically adaptive to observed time series.

- **Unified and scalable optimization** (see Sections 4.1–4.2). Reparameterizing acyclicity with a learnable permutation matrix enables joint optimization where both acyclicity enforcement and causal effect estimation are handled in a single stage. This avoids constrained optimization and the need for multi-stage procedures used in existing methods, and allows gradient-based optimizers to be applied directly to the entire learning problem, leading to improved efficiency and scalability for large-scale settings.

- **Extensive evaluation on real-world data** (see Section 5). We evaluate on three real-world benchmarks covering 11 datasets: IT monitoring (Aït-Bachir et al., 2023), SWaT (Maiti et al., 2023), and CausalRiver (Gideon et al., 2025). Across all datasets, our method achieves the best overall performance compared with 12 baselines in both discovery accuracy and efficiency. On CausalRiver, the largest benchmark to date, our method achieves substantially higher accuracy with over 6× speedup compared to competing methods, demonstrating strong scalability to high-dimensional data.

## 2 RELATED WORK

Many methods have been developed for time-series causal discovery (Gong et al., 2024). Some approaches ignore instantaneous dependencies and recover only lagged relationships, most notably Granger causality (Granger, 1969), which relies on VAR model (Sims, 1980). Examples include MVGC (Barrett et al., 2010; Barnett & Seth, 2014), TCDF (Nauta et al., 2019), and neural Granger causality (Tank et al., 2021). Constraint-based methods, in contrast, do not enforce acyclicity on instantaneous effects explicitly but rather handle it implicitly through conditional independence tests, such as tsFCI (Entner & Hoyer, 2010; Gerhardus & Runge, 2020), PCMCI+ (Runge et al., 2019b; Runge, 2020), and PCGCE (Assaad et al., 2022a). Since our focus is time-series causal discovery with instantaneous effects for theoretical completeness (following the argument of Assaad et al. (2022b) that true causal discovery should consider both instantaneous and lagged effects), we do not review above methods in detail, but include them as baselines in our experiments for comparison.

Our main focus is on time-series causal discovery methods that explicitly address the enforcement of acyclicity on instantaneous effects, most of which extend static causal discovery to temporal settings. Broadly, two families can be distinguished depending on how acyclicity is enforced.

The first family relies on discrete combinatorial search for a fixed causal order, followed by causal-effect estimation. This includes two noise-based methods. First, VARLiNGAM (Hyvärinen et al., 2010) extends LiNGAM (Shimizu et al., 2011) by combining a VAR with a non-Gaussian instantaneous model. Identifiability is achieved under the assumption of non-Gaussian errors (Shimizu et al., 2006), with acyclicity enforced through Independent Component Analysis (ICA) (Lee, 1998). Second, TiMINo (Peters et al., 2013) generalizes structural equation models to time series, using nonlinear additive noise models (Mooij et al., 2009) and residual-independence tests to iteratively identify sources and build an acyclic order. These methods are multi-stage, sensitive to order-estimation errors, and can suffer from escalating computational cost in high dimensions.

The second family enforces acyclicity by formulating it as an algebraic constraint within a constrained optimization problem. Proposed as a score-based method, DYNOTEARS (Pamfil et al., 2020) adapts NOTEARS (Zheng et al., 2018) to dynamic settings, jointly estimating causal effects within an SVAR formulation. It imposes acyclicity via the smooth equality constraint $h(W) = \text{tr}(e^{W \circ W}) - d = 0$, where $W$ is the weighted instantaneous effects and $d$ is the number of variables. This constraint is enforced through a complex augmented Lagrangian method (Nemirovsky, 1999). While this allows joint estimation of intra- and inter-slice effects, the augmented Lagrangian iterations are computationally demanding and unpredictable in large-scale settings.

In response to the limitations of existing acyclicity handling methods, we propose a new time-series causal discovery approach built on differentiable permutation learning (Maddison et al., 2017; Jang et al., 2017; Mena et al., 2018), particularly the Gumbel–Sinkhorn relaxation. Recent work has also applied differentiable permutations to static DAG learning (Charpentier et al., 2022; Chevalley et al., 2024; Annadani et al., 2023), but these methods operate in the i.i.d. setting and focus on learning a static causal ordering. In contrast, our formulation targets the time-series challenge of jointly estimating instantaneous and lagged effects while satisfying instantaneous acyclicity. Here the permutation serves not as a standalone objective but as a reparameterization that enables unified optimization over both structures, preserving acyclicity throughout training and avoiding the multi-stage pipelines used in prior approaches.

## 3 PROBLEM SETUP

While the objective of time-series causal discovery is to construct a valid causal graph that captures both instantaneous and lagged dependencies among variables, the SVAR model reframes this task as estimating the model coefficients that best explain the observed data, since each coefficient matrix $B_\tau$ corresponds to a subgraph. Formally, the SVAR model is given by:

$$\mathbf{x}_t = \sum_{\tau=0}^{k} B_\tau \mathbf{x}_{t-\tau} + \boldsymbol{\epsilon}_t, \tag{1}$$

where $\mathbf{x}_t \in \mathbb{R}^d$ is the vector of observed variables at time $t$. $k$ is the maximum lag order considered. $B_\tau \in \mathbb{R}^{d \times d}$ is the coefficient matrix at lag $\tau$: $B_0$ encodes instantaneous effects, while $B_\tau$ for $\tau > 0$ encodes lagged effects. $\boldsymbol{\epsilon}_t$ is the vector of error terms at time $t$.

The linear structure of SVAR offers strong interpretability. Its coefficients not only reveal the presence of causal relationships but also quantify their strength and direction (positive or negative). This provides richer information than methods that merely identify links and further enables direct modeling of system dynamics. For this reason, we adopt the linear SVAR model as our foundation.

**Identifiability** Identifiability is a central issue in SVAR models (Pamfil et al., 2020). For lagged effects, identification follows from standard VAR assumptions. In contrast, instantaneous effects are harder, since covariance information alone is insufficient to uniquely identify $B_0$ (Hyvärinen et al., 2010). Two common sufficient conditions under independent $\boldsymbol{\epsilon}_t$ are: (i) non-Gaussian noise, yielding identifiability via ICA/Marcinkiewicz arguments (Hyvärinen et al., 2010; Lanne et al., 2017); and (ii) Gaussian noise with equal error variances (e.g., standard Gaussian), under which the DAG is identifiable, together with acyclicity of the instantaneous graph (Peters & Bühlmann, 2014). Following Pamfil et al. (2020), we assume at least one of these conditions holds. In line with standard linear SVAR formulations, we additionally assume causal sufficiency and faithfulness, the same assumptions made by VARLiNGAM and DYNOTEARS.

## 4 METHODOLOGY

### 4.1 UNIFIED OPTIMIZATION

As discussed in Section 3, fitting the SVAR model reduces to estimating coefficient matrices that best explain the observed data. We thus define our optimization objective $l(\tilde{B}_0, \{B_\tau\})$ as the mean squared error (MSE) of the SVAR residuals:

$$\min_{\tilde{B}_0, \{B_\tau\}_{\tau=1}^k} l(\tilde{B}_0, \{B_\tau\}) = \frac{1}{N} \sum_{t=k+1}^{T} \left\| (I - \tilde{B}_0)\mathbf{x}_t - \sum_{\tau=1}^{k} B_\tau \mathbf{x}_{t-\tau} \right\|_2^2, \qquad (2)$$

where $T$ is the total number of time points, $N = T - k$ is the number of effective samples, and $\|\cdot\|_2^2$ denotes the squared $L_2$-norm. Crucially, the optimization is performed over $\tilde{B}_0$, the instantaneous effects matrix constrained to be acyclic. This matrix is derived from an unconstrained matrix $B_0$ with the Gumbel–Sinkhorn technique (detailed in Section 4.3).

Since many real-world causal graphs exhibit moderate sparsity, $L_1$ regularization and its variants are commonly employed as sparsity penalties (Hyvärinen et al., 2010). We adopt standard $L_1$ regularization, consistent with prior work (Pamfil et al., 2020). The final penalized objective is:

$$\min_{\tilde{B}_0, \{B_\tau\}_{\tau=1}^k} f(\tilde{B}_0, \{B_\tau\}) = l(\tilde{B}_0, \{B_\tau\}) + \lambda_0 \|\tilde{B}_0\|_1 + \lambda_\tau \sum_{\tau=1}^{k} \|B_\tau\|_1, \qquad (3)$$

where $\|\cdot\|_1$ is the element-wise $L_1$-norm and $\lambda_0, \lambda_\tau$ are penalty hyperparameters.

**Direct gradient-based optimization** Though this form of optimization objective has been applied to causal discovery before (Pamfil et al., 2020), it has so far been used as constrained optimization solved with nested updates, which makes it impossible to apply a direct gradient-based optimizer to the entire problem. Our method, however, reparameterizes the acyclicity enforcement (see Section 4.3), resolving this issue and relaxing the problem into a unified and unconstrained optimization. This enables direct gradient-based optimization of the whole objective, which is theoretically more efficient than constrained approaches in this setting (Nocedal & Wright, 2006; Jaggi, 2013).

### 4.2 THEORETICAL JUSTIFICATION

The validity of the objective in equation 2 follows from standard properties of least-squares estimation in linear structural models. Under the exogeneity and finite-moment assumptions of the SVAR formulation, the population squared-loss risk is minimized at the true coefficient matrices, irrespective of the distribution of the innovations. Thus, while squared loss coincides with the Gaussian maximum-likelihood estimator, it more generally functions as a quasi-likelihood estimator whose consistency does not rely on Gaussian noise. Appendix A.3 provides a formal argument establishing this result in our setting.

The full penalized objective in equation 3 is supported by the analysis of Aragam et al. (2015), who show that penalized least-squares estimators with concave penalties achieve uniform support recovery and deviation bounds when $n \gtrsim d \log p$, and that their global minimizers remain statistically consistent under a beta-min condition. These results justify the use of $\ell_1$ penalties for sparsity control. Although their guarantees are derived under Gaussian designs, empirical studies (Pamfil et al., 2020) and our consistency argument for the unpenalized loss demonstrate that squared-loss–based estimation remains effective beyond the Gaussian setting. Consequently, minimizing residual error subject to the structural constraints encoded by our permutation-based acyclicity enforcement provides a statistically grounded approach to estimating the SVAR coefficients.

### 4.3 ACYCLICITY ENFORCEMENT VIA DIFFERENTIABLE PERMUTATION

The acyclicity of $B_0$ is satisfied if there exists a permutation matrix $P$ such that $PB_0P^\top$ is (close to) strictly lower triangular (Hyvärinen et al., 2010). Thus, enforcing acyclicity on instantaneous effects can be framed as finding a permutation that triangularizes $B_0$ while still fitting the data well. However, permutation matrices are discrete and therefore incompatible with the continuous optimization

in equation 3. The key challenge is to integrate this combinatorial choice into a differentiable framework. To address this, we employ the Gumbel–Sinkhorn technique (Mena et al., 2018), which learns a continuous parameterization that yields a differentiable approximation of $P$.

### 4.3.1 Gumbel–Sinkhorn Relaxation for Causal Ordering

The Gumbel–Sinkhorn method provides a differentiable relaxation of permutation matrices, enabling gradients to backpropagate through them (Mena et al., 2018). We develop a Gumbel–Sinkhorn method to learn the causal order as follows: we first introduce a learnable matrix of unconstrained real-valued logits $\Lambda \in \mathbb{R}^{d \times d}$, which encodes the model's preference over variable orderings. To encourage exploration during training, we apply the "Gumbel trick" by adding a noise matrix $\mathbf{G}$ with i.i.d. Gumbel entries, forming perturbed scores $\Lambda + \mathbf{G}$. The perturbed logits are then temperature-scaled:

$$X = \frac{\Lambda + \mathbf{G}}{\tau_{\text{temp}}}, \tag{4}$$

where $\tau_{\text{temp}}$ is a temperature parameter: higher values yield smoother distributions for stable early training, while lower values sharpen the matrix toward a discrete permutation.

We then apply the Sinkhorn operator $\mathcal{S}(\cdot)$, which iteratively normalizes the rows and columns of a positive matrix until convergence to a doubly stochastic matrix (where both rows and columns sum to 1), i.e. a continuous relaxation of a permutation. The procedure is defined as:

$$\mathcal{S}^0(X) = \exp(X), \tag{5}$$

$$\mathcal{S}^l(X) = \mathcal{T}_c(\mathcal{T}_r(\mathcal{S}^{l-1}(X))), \quad l \geq 1, \tag{6}$$

$$\mathcal{S}(X) = \lim_{l \to \infty} \mathcal{S}^l(X), \tag{7}$$

where exponentiation in $\mathcal{S}^0(X)$ ensures strictly positive entries, a requirement for the normalization steps, and $\mathcal{T}_r$, $\mathcal{T}_c$ denote row and column normalization, respectively. In practice, this limit is approximated with a fixed number of iterations $L$. The resulting soft permutation matrix $\tilde{P}$ is:

$$\tilde{P} = \mathcal{S}\left(\frac{\Lambda + \mathbf{G}}{\tau_{\text{temp}}}\right), \tag{8}$$

which converges to a discrete permutation as $\tau_{\text{temp}} \to 0$, but remains differentiable for $\tau_{\text{temp}} > 0$.

**Recovery of Hard Permutation** During training, the relaxed permutation $\tilde{P}$ remains differentiable, which allows gradients to flow through the optimization. However, causal discovery requires a hard causal ordering. Therefore, we recover a hard permutation matrix by projecting $\tilde{P}$ onto the nearest discrete permutation. In practice, this can be done by taking the row- and column-wise $\arg\max$ after the final Sinkhorn iteration, yielding a valid $P$. This ensures that the final learned causal order corresponds to a discrete DAG structure, while the relaxation only serves to enable backpropagation during optimization.

### 4.3.2 Enforcing Acyclicity

The differentiable permutation matrix $\tilde{P}$ enables acyclicity enforcement on $B_0$ directly within the optimization loop. Starting from the unconstrained instantaneous effects $B_0$, we first permute into the learned causal order:

$$B_0' = \tilde{P} B_0 \tilde{P}^\top. \tag{9}$$

Then apply a strictly lower-triangular mask:

$$B_0'' = \text{tril}(B_0', -1), \tag{10}$$

which zeros the diagonal and upper-triangular entries to enforce acyclicity in the permuted space.

Finally, map back to the original variable order,

$$\tilde{B}_0 = \tilde{P}^\top B_0'' \tilde{P}, \tag{11}$$

which yields the acyclicity-constrained instantaneous matrix $\tilde{B}_0$ used in the objective $f(\tilde{B}_0, \{B_\tau\})$ (Eq. 3). By embedding these operations in the forward pass, the DAG constraint becomes fully differentiable, enabling true end-to-end learning of the causal structure.

## 5 EVALUATION

### 5.1 EXPERIMENTAL SETTINGS

#### 5.1.1 DATASETS

As noted by Reisach et al. (2021), synthetic data is easy to game and may not reflect a method's true performance. We therefore exclude synthetic benchmarks and focus on three real-world benchmarks: the small-scale IT Monitoring Aıt-Bachir et al. (2023), the medium-scale SWaT Maiti et al. (2023), and the large-scale CausalRiver Gideon et al. (2025). An overview is provided in Table 1.

Table 1: Overview of benchmarks and datasets used in our experiments. SR stands for sampling rate, #D stands for number of time series data, and #V stands for number of variables.

| Benchmark | Origin | Dataset | Scenario | SR | #D | #V |
|---|---|---|---|---|---|---|
| IT Monitoring | UAI2023 | MOM 1 | Message-Oriented Middleware | 1 sec | 288 | 7 |
| | | MOM 2 | | | 364 | 7 |
| | | Storm | Storm ingestion topology | 1 min | 991 | 8 |
| | | Web 1 | Web server activity | 5 mins | 7500 | 10 |
| | | Web 2 | | | 7501 | 10 |
| | | AntiV 1 | Antivirus impact on server | 5 mins | 1320 | 13 |
| | | AntiV 2 | | | 1321 | 13 |
| SWaT | Arxiv2023 | SWaT | Water treatment | 1 sec | ≈97k | 51 |
| CausalRiver | ICLR2025 | Flood | River discharge (Elbe Flood) | 15 min | 3010 | 42 |
| | | Bavaria | River discharge (Bavaria) | 15 min | ≈175k | 494 |
| | | Germany | River discharge (East Germany) | 15 min | ≈175k | 666 |

#### 5.1.2 BASELINES

For comparison, we use 12 time-series causal discovery methods as baselines, summarized in Table 2, selected for their relevance and prevalence in prior studies. Although VAR is not a strict causal discovery method, we include it as a classical and widely adopted baseline. Most baselines do not explicitly handle instantaneous effects (see Section 2); notable exceptions are DYNOTEARS (Pamfil et al., 2020), VARLiNGAM (Hyvärinen et al., 2010), and TiMINo (Peters et al., 2013), which are directly comparable to our method and are of particular interest. We also include TCDF (Nauta et al., 2019) and three neural Granger causality variants (Tank et al., 2021) as representatives of recent deep learning–based approaches for nonlinear, high-dimensional causal discovery.

Table 2: Baselines used in our experiments.

| Type | Methods |
|---|---|
| Traditional | VAR (Sims, 1980), tsFCI (Entner & Hoyer, 2010), VARLiNGAM (Hyvärinen et al., 2010), TiMINo (Peters et al., 2013), MVGC (Barnett & Seth, 2014) |
| State-of-the-art | TCDF (Nauta et al., 2019), DYNOTEARS (Pamfil et al., 2020), PCMCI+ Runge (2020), Neural Granger causality variants (cMLP, cLSTM, cRNN) (Tank et al., 2021), PCGCE Assaad et al. (2022a) |

#### 5.1.3 METRICS

We evaluate performance using Precision, Recall, and F1-Score, following prior work Assaad et al. (2022b); Aıt-Bachir et al. (2023); Nauta et al. (2019), and report F1-Score in this section. While some studies use AUC-ROC (Gideon et al., 2025), this metric can remain high under severe class imbalance, which is a common issue in causal discovery. In contrast, F1-Score is stricter and directly balances false positives and false negatives, making it more suitable for evaluation of this task.

### 5.1.4 PARAMETER SETTING

Experiments were conducted on an Intel Core i7-14700K (20 cores, 28 threads, 3.4 GHz, 33 MB cache) and 128GB of DDR5-5600 memory. The CPU was used for fair comparison, as some baselines only support CPU execution. Runtimes were limited to 3 hours per run Guo & Luk (2022); longer runs are reported as TLE (Time Limit Exceeded). Baselines were executed with default parameters. Maximum lags of 3, 5, 10, and 15 were used as in Aıt-Bachir et al. (2023), excluding TCDF which infers lags automatically. VARLiNGAM applied its built-in pruning, while DYNOTEARS, VAR, and our method pruned coefficients below 0.01 Assaad et al. (2022b); Aıt-Bachir et al. (2023). Sparsity penalties $\lambda_{B_0}$ and $\lambda_{B_\tau}$ were fixed at 0.001 Pamfil et al. (2020). Models were trained using Adam ($lr = 0.002$) for a maximum of 6000 epochs with early stopping, with 20 Sinkhorn iterations and a temperature setting of $\tau_{\text{temp}} = 0.01$ Mena et al. (2018).

### 5.2 PERFORMANCE ANALYSIS

In this section, we report the F1 scores of all methods across three benchmarks. The results on the IT Monitoring benchmark with a maximum lag of 3 are shown in Table 3, while the results on the SWaT and CausalRiver benchmarks with maximum lags of 5 and 10 are presented in Table 4. Additional F1 score results for other lag settings are provided in Appendix A.4, with the corresponding Recall and Precision results given in Appendix A.6 and Appendix A.7, respectively.

Table 3: F1 scores on IT Monitoring with a maximum lag of 3. For each dataset, the highest and second-highest scores are highlighted in dark green and light green, respectively. The last two columns report the frequency with which each method achieves the highest F1 score (wins, W) and the second-highest score (runner-ups, R), with the highest frequency also marked in light green.

| Method | MoM 1 | MoM 2 | Ingestion | Web 1 | Web 2 | AntiV 1 | AntiV 2 | W | R |
|---|---|---|---|---|---|---|---|---|---|
| VAR | 0.1667 | 0.1622 | 0.2449 | 0.2222 | 0.2895 | 0.2047 | 0.2188 | 0 | 1 |
| MVGC | 0.0909 | 0.0000 | 0.0667 | 0.2059 | 0.2400 | 0.1613 | 0.1374 | 0 | 0 |
| cLSTM | 0.3390 | 0.3390 | 0.2500 | 0.2321 | 0.2500 | 0.1739 | 0.1739 | 0 | 2 |
| cMLP | 0.2857 | 0.1622 | 0.1379 | 0.1905 | 0.1081 | 0.2247 | 0.2857 | 0 | 0 |
| cRNN | 0.1714 | 0.1500 | 0.2069 | 0.1778 | 0.1667 | 0.2273 | 0.2903 | 2 | 0 |
| TCDF | 0.0000 | 0.0000 | 0.0000 | 0.0000 | 0.1053 | 0.1818 | 0.2000 | 0 | 0 |
| PCMCI+ | 0.0000 | 0.0000 | 0.0000 | 0.3243 | 0.1579 | 0.0357 | 0.0833 | 1 | 0 |
| tsFCI | 0.2286 | 0.0870 | 0.0000 | 0.1818 | 0.1714 | 0.1852 | 0.1159 | 0 | 0 |
| PCGCE | 0.0909 | 0.1538 | 0.1935 | 0.2143 | 0.1961 | 0.2250 | 0.2400 | 0 | 2 |
| DYNOTEARS | 0.2857 | 0.2353 | 0.1538 | 0.2623 | 0.2895 | 0.1905 | 0.2056 | 0 | 2 |
| VARLiNGAM | 0.0000 | 0.0000 | 0.3846 | 0.2593 | 0.2667 | 0.1923 | 0.2188 | 1 | 0 |
| TiMINo | 0.1538 | 0.1818 | 0.0000 | 0.0000 | 0.0000 | 0.0000 | 0.0000 | 0 | 0 |
| Our Method | 0.4000 | 0.3415 | 0.2769 | 0.3243 | 0.3243 | 0.1830 | 0.1875 | 4 | 1 |

Table 4: F1 scores on SWaT and CausalRiver with maximum lags of 5 and 10. The last two columns report the counts of wins (W) and runner-ups (R). TLE: Time Limit (3 hours) Exceeded.

| Method | Lag = 5 | | | | Lag = 10 | | | | W | R |
|---|---|---|---|---|---|---|---|---|---|---|
| | SWaT | Flood | Bavaria | Germany | SWaT | Flood | Bavaria | Germany | | |
| VAR | 0.0565 | 0.0607 | 0.0059 | 0.0045 | 0.0539 | 0.0574 | 0.0056 | 0.0041 | 0 | 4 |
| MVGC | 0.0767 | 0.0559 | TLE | TLE | 0.0787 | 0.0572 | TLE | TLE | 0 | 0 |
| cLSTM | 0.0279 | 0.0465 | TLE | TLE | 0.0000 | 0.0466 | TLE | TLE | 0 | 0 |
| cMLP | 0.0902 | 0.1203 | TLE | TLE | 0.0833 | 0.1158 | TLE | TLE | 0 | 2 |
| cRNN | 0.0788 | 0.0952 | TLE | TLE | 0.0729 | 0.1009 | TLE | TLE | 0 | 0 |
| TCDF | 0.0000 | 0.0000 | TLE | TLE | 0.0000 | 0.0000 | TLE | TLE | 0 | 0 |
| PCMCI+ | TLE | TLE | TLE | TLE | TLE | TLE | TLE | TLE | 0 | 0 |
| tsFCI | TLE | TLE | TLE | TLE | TLE | TLE | TLE | TLE | 0 | 0 |
| PCGCE | 0.0648 | 0.1116 | TLE | TLE | 0.0827 | 0.0813 | TLE | TLE | 0 | 0 |
| DYNOTEARS | 0.0206 | 0.1342 | TLE | TLE | 0.0215 | 0.1333 | TLE | TLE | 0 | 2 |
| VARLiNGAM | 0.0660 | 0.0403 | TLE | TLE | 0.0621 | 0.0362 | TLE | TLE | 0 | 0 |
| TiMINo | 0.0580 | 0.0000 | TLE | TLE | 0.0435 | 0.0000 | TLE | TLE | 0 | 0 |
| Our Method | 0.2202 | 0.3000 | 0.1751 | 0.1351 | 0.2182 | 0.3226 | 0.1860 | 0.1388 | 8 | 0 |

On the small-scale IT Monitoring datasets (Table 3), our method achieves the highest F1 score on four out of seven datasets and remains competitive on Ingestion, yielding the largest overall count of wins and runner-ups (5/7). This demonstrates consistent performance, whereas other methods show strong dataset-specific biases, for example, VARLiNGAM excels on Ingestion but fails on MoM1/2, DYNOTEARS performs well on Web datasets, and cRNN and PCGCE on AntiVirus. None, however, achieve robust performance across most datasets as our method does.

On AntiV1/AntiV2, our method performs less strongly. This appears related to the sensitivity of pruning thresholds on these antivirus datasets, whose sparse and irregular dynamics arise from mixed sampling rates, partial sleeping series, and interpolation steps (Aıt-Bachir et al., 2023). Under such conditions, a fixed threshold can be fragile: small but meaningful effects become hard to distinguish from noise, while bursty fluctuations may be retained. As shown in Appendix A.5, tuning the threshold improves results, which indicates a methodological limitation.

On the medium-scale SWaT and Flood datasets and the large-scale Bavaria and Germany datasets, our method shows a clear advantage. In all eight cases in Table 4, it achieves the best performance, often with F1 scores more than double those of the second-best method. This dominance persists across other lag settings (Appendix A.4). On large-scale datasets, our method consistently finishes within the 3-hour time limit—while most baselines fail to complete—and still attains the highest F1 scores. Performance remains stable across lag values, indicating that even when lag doubles, our method captures key causal links without introducing excessive spurious ones.

VARLiNGAM, DYNOTEARS and TiMINo are the closest and most directly comparable baselines, as they also address time-series causal discovery with instantaneous acyclicity. Across datasets and lag settings, our method consistently outperforms. This suggests that our approach of jointly estimating instantaneous and lagged effects through a unified optimization may yield more reliable recovery of causal structures and help mitigate the error propagation inherent in multi-stage pipelines.

## 5.3 COMPUTATIONAL EFFICIENCY

In this section, we report the runtime (in seconds) of each method on all datasets with maximum lag set to 15, as shown in Table 5, where the fastest and second-fastest methods are highlighted. Results for lags 3, 5, and 10 are provided in Appendix A.8. Although VAR is included as a baseline for accuracy, we exclude it in this section. Unlike other methods, which aim at causal discovery (with or without explicit acyclicity enforcement), VAR doesn't attempt to discover the true causal structure; it only encodes dependencies over time. Its consistently shorter runtimes therefore reflect solving a simpler problem, and including them would give a misleading impression of efficiency.

Table 5: Runtime (in seconds) with maximum lag of 15. TLE: Time Limit (3 hours) Exceeded. VAR is excluded from comparison because it does not consider instantaneous causal effects.

| Method | MoM 1 | MoM 2 | Storm | Web 1 | Web 2 | AntiV 1 | AntiV 2 | SWaT | Flood | Bavaria | Germany |
|---|---|---|---|---|---|---|---|---|---|---|---|
| VAR (Ref. Only) | 0.02 | 0.03 | 0.03 | 0.06 | 0.06 | 0.06 | 0.07 | 0.79 | 0.60 | 82.69 | 151.38 |
| MVGC | 0.30 | 0.36 | 0.24 | 2.00 | 3.48 | 10.35 | 10.75 | 3632.96 | 4842.76 | TLE | TLE |
| cLSTM | 32.09 | 39.31 | 61.60 | 199.44 | 221.08 | 131.27 | 125.94 | TLE | 727.86 | TLE | TLE |
| cMLP | 12.45 | 12.58 | 14.55 | 43.48 | 54.03 | 35.85 | 35.80 | 836.62 | 277.24 | TLE | TLE |
| cRNN | 22.15 | 24.26 | 34.21 | 216.56 | 212.54 | 107.78 | 117.74 | 6933.72 | 1612.64 | TLE | TLE |
| TCDF | 8.69 | 9.02 | 9.94 | 58.36 | 58.55 | 16.65 | 17.81 | 550.15 | 258.03 | TLE | TLE |
| PCMCI+ | 0.51 | 0.89 | 4.50 | 145.98 | 61.36 | 34.09 | 38.51 | TLE | TLE | TLE | TLE |
| tsFCI | 1252.319 | 1232.84 | 2.70 | 4649.73 | TLE | TLE | TLE | TLE | TLE | TLE | TLE |
| PCGCE | 0.33 | 0.68 | 1.48 | 10.10 | 5.62 | 16.41 | 8.78 | 725.87 | 86.29 | TLE | TLE |
| DYNOTEARS | 8.36 | 4.92 | 0.39 | 64.06 | 403.92 | 85.21 | 490.66 | 4641.99 | 795.53 | TLE | TLE |
| VARLiNGAM | 0.35 | 0.38 | 0.69 | 9.23 | 9.60 | 2.22 | 2.14 | 5077.79 | 54.87 | TLE | TLE |
| TiMINo | 1.34 | 1.99 | 2.51 | 16.68 | 12.77 | 8.23 | 6.22 | 770.68 | 91.03 | TLE | TLE |
| Our Method | 7.25 | 5.62 | 6.99 | 9.97 | 13.99 | 7.54 | 8.56 | 64.29 | 20.50 | 1915.20 | 3240.05 |

On the first seven small-scale datasets (from MoM 1 to AntiV 2), our method is not the fastest. Traditional methods such as VAR, MVGC, and VARLiNGAM have lower runtimes. However, our method consistently completes its analysis in under 15 seconds in all cases. This performance is highly competitive and often faster than many complex models like cLSTM, cRNN, and tsFCI. In practical applications, the minor difference of a few seconds is negligible.

The true advantages of our method become obvious when applied to medium and large-scale datasets. On the SWaT dataset, our method is over 8 times faster than the next-best competitor, TCDF. A direct comparison with VARLiNGAM and DYNOTEARS, and TiMINo is most telling. On SWaT, our algorithm is approximately 79 times faster than VARLiNGAM, 72 times faster than DYNOTEARS, and 12 times faster than TiMINo. This performance gap is also evident on the Flood dataset, where our method is nearly 39 times faster than DYNOTEARS.

The most compelling evidence of scalability is observed on the large-scale Bavaria and Germany datasets. On these complex, high-dimensional tasks, all competing methods failed to produce a result within the 3-hour time limit, while our method completed the analysis in approximately 32 minutes for Bavaria and 54 minutes for Germany. This indicates that our method is at least 6 times faster on Bavaria and 3 times faster on Germany than the strongest alternatives.

The substantial gap also reflects the advantage of our unified optimization, in contrast to VAR-LiNGAM, DYNOTEARS, and TiMINo, which rely on multi-stage pipelines to enforce instantaneous acyclicity and estimate lagged effects. Additional comparisons with GPU-accelerated baselines in Appendix A.9 confirm that this efficiency advantage remains even with GPU acceleration.

## 6 Discussion

**Limitations** Our work focuses on enforcing acyclicity on instantaneous effects and demonstrates improved accuracy and efficiency in Section 5, but several limitations remain. We did not analyze the convergence or potential sub-optimality of the Gumbel–Sinkhorn relaxation beyond empirical evidence. Our study of lag choices was limited, and a more systematic exploration may be valuable. The use of $L_1$ regularization is a practical choice for structure recovery rather than an assumption of universal sparsity; denser systems may require stronger regularization or alternative priors. We acknowledge these limitations and leave deeper investigation to future work.

**Causality and Instantaneous Effects** The instantaneous effects in our SVAR formulation reflect dependencies at the measurement timescale, not necessarily true system-level interactions. Many physical processes propagate over nonzero $\varepsilon$-time, so with sufficiently fine sampling contemporaneous edges vanish and only lagged links remain. Prior work on subsampling and timescale mismatch shows that apparent instantaneous edges at coarse sampling can be artifacts of temporal aggregation (Gong et al., 2015; Hyttinen et al., 2017; Tank et al., 2019; Liu et al., 2023; Abavisani et al., 2022). Our method therefore recovers structure at the observed resolution; when sampling is coarse, some instantaneous effects may reflect aggregated lagged influences. Addressing system-timescale recovery or sampling-rate mismatch is an important direction for future work.

## 7 Conclusion

A major challenge in time-series causal discovery is enforcing acyclicity on instantaneous effects, which most methods treat as a hard constraint, limiting flexibility and increasing computational cost. We address this with a new SVAR-based method that uses the Gumbel–Sinkhorn technique to impose acyclicity via a soft, differentiable permutation, enabling unified and efficient end-to-end optimization with gradient-based optimizer. Evaluations on IT monitoring, SWaT, and CausalRiver benchmarks show that our approach is both accurate and scalable: it delivers competitive and stable results on seven smaller IT monitoring datasets, achieves up to twice the F1 score of the second-best method on larger datasets, and provides substantial speedups, reaching up to 72× on SWaT and over 6× on CausalRiver compared to methods with explicit acyclicity constraints.

**Future Work** Several directions remain for future exploration. First, additional experiments could be conducted to address the limitations discussed in Section 6. Second, as discussed in Section 5.2, results can be sensitive to the choice of fixed pruning threshold, which motivates adaptive or stability-based pruning strategies that better distinguish noise from weak but meaningful effects. Another direction is to extend the model to capture non-linear causal relationships by incorporating non-linear functions or kernels, since our acyclicity enforcement mechanism is model-agnostic. Finally, as noted in Section 4.1, we use simple $L_1$ penalties for sparsity control, but strategies such as progressively increasing penalties across lags can be explored to further regularize causal discovery.

# 8 REPRODUCIBILITY STATEMENT

All hyperparameter settings are listed in Section 5.1. Details on preprocessing and postprocessing are provided in Appendix A.1 and Appendix A.2, respectively.

Our code is anonymously available at `https://anonymous.4open.science/r/Time-Series-Causal-Discovery-via-Differentiable-Permutations-6170/`.

The IT Monitoring and CausalRiver benchmarks are publicly accessible. The IT Monitoring benchmark can be obtained from `https://github.com/ckassaad/Case_Studies_of_Causal_Discovery_from_IT_Monitoring_Time_Series`, and the CausalRiver benchmark from `https://github.com/CausalRivers/causalrivers`. The SWaT dataset is private and accessible only under agreement; we obtained permission to use it in this study.

For the baselines used in our experiments, all implementations are in Python:

- VAR: available in the `statsmodels` library.
- MVGC: implemented by us in Python.
- Neural Granger causality variants (cLSTM, cRNN, cMLP): `https://github.com/iancovert/Neural-GC`.
- TCDF: `https://github.com/M-Nauta/TCDF`.
- PCMCI+: `https://github.com/jakobrunge/tigramite`.
- tsFCI: `https://sites.google.com/site/dorisentner/publications/tsfci` (originally in R; we provide a Python reimplementation).
- PCGCE: `https://github.com/ckassaad/PCGCE`.
- DYNOTEARS: available in the `causalnex` library.
- VARLiNGAM: `https://github.com/cdt15/lingam`.
- TiMINo: `https://web.math.ku.dk/~peters/code.html` (originally in R; we provide a Python reimplementation).

All experiments can be reproduced using the above code and datasets.

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

## USE OF LARGE LANGUAGE MODELS (LLMS)

Large Language Models (LLMs) were used in this work solely as writing assistant. Specifically, they were employed to refine grammar, improve clarity, shorten text when requested, and polish the presentation of results (e.g., experimental analysis, future work discussion, and table overviews). In addition, LLMs were used to help organize code and improve comments, but not for developing or directly writing code implementation for the main method.

LLMs were not used for research ideation, methodological design, data analysis, or result interpretation. All conceptual contributions, experimental design choices, data processing, and analysis were conducted entirely by the authors. The authors take full responsibility for the contents of this paper.

## A  APPENDIX

### A.1  PREPROCESSING

For all datasets, we preprocess the raw data by removing timestamps to ensure fair comparison across methods. Since the ground-truth causal graphs are provided in different formats across benchmarks, we convert them into a standardized summary matrix in `.npy` format for evaluation.

For the SWaT dataset (Maiti et al., 2023), we resample the data at 5-second intervals instead of the original 1-second sampling rate. This is because system responses in SWaT typically exhibit delays of more than 10 seconds. Without resampling, the valid maximum lag would be unrealistically large, making evaluation infeasible and potentially producing invalid results. In addition, we remove variables that remain constant throughout the dataset, since they provide no statistical information and can cause certain methods (e.g., VARLiNGAM) to crash.

### A.2  POSTPROCESSING

All ground-truth graphs are represented as *summary graphs* (see definition in Assaad et al. (2022b)). Accordingly, we process the outputs of all methods (including VAR, VARLiNGAM, and our method), which generate separate results for different lags, into a single summary graph. Specifically, for each edge position we take the maximum value across lags, including instantaneous effects. Following Assaad et al. (2022b); Aıt-Bachir et al. (2023); Gideon et al. (2025), we remove diagonal elements when evaluating, as they represent self-autocorrelation, which—although important—does not provide additional information about causality between variables. Finally, for methods requiring a pruning threshold, we apply a fixed threshold of 0.01, consistent with prior work (Assaad et al., 2022b; Aıt-Bachir et al., 2023).

## A.3 CONSISTENCY OF THE PENALIZED LEAST-SQUARES ESTIMATOR

This section provides a formal justification for using the penalized mean squared error (MSE) objective in our method, even when identifiability of the instantaneous structure relies on *non-Gaussian* noise. Our goal is to clarify that the squared-loss objective is best interpreted as a *quasi-likelihood* or *penalized least-squares* estimator, which remains statistically consistent under broad noise distributions and does not require Gaussianity.

We consider the Structural VAR (SVAR) model:

$$x_t = \sum_{\tau=0}^{k} B_\tau^\star x_{t-\tau} + \epsilon_t, \tag{12}$$

where $\epsilon_t$ is a vector of independent innovations with zero mean. The instantaneous matrix $B_0^\star$ is assumed lower-triangular under the (unknown) causal order, ensuring recursive solvability of equation 12.

**Assumptions.** We make the following standard assumptions:

(A1) (**Stationarity and ergodicity**) $\{x_t\}$ is covariance-stationary and ergodic.

(A2) (**Orthogonality / exogeneity**) After ordering variables according to the true causal permutation, the SVAR equations are recursive; hence each innovation satisfies

$$\mathbb{E}[\epsilon_{t,j}\, r_{t,j}^\top] = 0,$$

where $r_{t,j}$ collects regressors of equation $j$: all lagged variables $\{x_{t-\tau}\}_{\tau=1}^{k}$ and the previously ordered contemporaneous variables. For notational simplicity, we write this vector generically as $z_t$ and express the condition succinctly as

$$\mathbb{E}[\epsilon_t z_t^\top] = 0.$$

(A3) (**Finite second moments**) $\mathbb{E}\|x_t\|^2 < \infty$ and $\mathbb{E}\|\epsilon_t\|^2 < \infty$.

(A4) (**Full-rank regressor covariance**) The covariance matrix $\Sigma_z = \mathbb{E}[z_t z_t^\top]$ is positive definite.

Note that none of the assumptions above require $\epsilon_t$ to be Gaussian. Assumption (A2) is the standard exogeneity condition enabling identification of linear structural parameters, and it is satisfied automatically in the correctly ordered SVAR due to its recursive structure.

POPULATION RISK MINIMIZER

Define the population MSE:

$$R(\theta) = \mathbb{E}\Big\|x_t - \sum_{\tau=0}^{k} B_\tau x_{t-\tau}\Big\|_2^2, \quad \theta = \{B_\tau\}_{\tau=0}^{k}. \tag{13}$$

Substituting the true model equation 12 gives

$$x_t - \sum_{\tau=0}^{k} B_\tau x_{t-\tau} = \epsilon_t + \sum_{\tau=0}^{k} (B_\tau^\star - B_\tau) x_{t-\tau}.$$

Expanding and applying (A2) removes the cross terms:

$$R(\theta) = \mathbb{E}\|\epsilon_t\|_2^2 + \mathbb{E}\Big\|\sum_{\tau=0}^{k} (B_\tau^\star - B_\tau) x_{t-\tau}\Big\|_2^2.$$

The second term is nonnegative and equals zero iff $B_\tau = B_\tau^\star$ for all $\tau$, by (A4). Thus:

**Lemma A.1** (Uniqueness of Population Minimizer). *Under (A1)–(A4), the unique minimizer of $R(\theta)$ is the true parameter $\theta^\star$.*

This shows that *the population squared-loss risk is minimized at the true SVAR coefficients regardless of the distribution of the noise*. Gaussianity is not required.

CONSISTENCY OF THE EMPIRICAL MINIMIZER

Define the empirical risk:

$$\widehat{R}_N(\theta) = \frac{1}{N} \sum_{t=k+1}^{T} \left\| x_t - \sum_{\tau=0}^{k} B_\tau x_{t-\tau} \right\|_2^2, \qquad N = T - k.$$

By ergodicity (A1) and finite second moments (A3),

$$\widehat{R}_N(\theta) \xrightarrow{a.s.} R(\theta) \quad \text{uniformly in compact sets.}$$

Combining this with Lemma A.1 yields a standard M-estimation result:

**Theorem A.2** (Consistency of Least-Squares Estimator). *Let $\widehat{\theta}_N = \arg\min_\theta \widehat{R}_N(\theta)$. Under (A1)–(A4),*

$$\widehat{\theta}_N \xrightarrow{p} \theta^\star.$$

CONSISTENCY UNDER $\ell_1$ REGULARIZATION

Our estimator minimizes the penalized objective:

$$\widehat{\theta}_N = \arg\min_\theta \left[ \widehat{R}_N(\theta) + \lambda_0 \|B_0\|_1 + \sum_{\tau=1}^{k} \lambda_\tau \|B_\tau\|_1 \right].$$

When $\lambda_\tau \to 0$ at an appropriate rate (e.g., $\lambda_\tau = o(1)$), penalized least-squares inherits the same consistency properties; see, for example, Aragam et al. (2015) for support-recovery and deviation bounds in penalized regression. Thus:

**Corollary A.3** (Consistency of Penalized Estimator). *Under (A1)–(A4), if $\lambda_\tau \to 0$ as $N \to \infty$, then*

$$\widehat{\theta}_N \xrightarrow{p} \theta^\star.$$

DISCUSSION

This analysis shows that the squared-loss objective used in our method is appropriately viewed as a quasi-likelihood estimator whose validity does not depend on Gaussian noise assumptions. Although identifiability of the instantaneous coefficient matrix $B_0^\star$ may rely either on non-Gaussian innovations or on equal-variance Gaussian noise, the consistency of the least-squares estimator remains guaranteed under both regimes due to the orthogonality and moment conditions imposed on the SVAR innovations. Consequently, employing the MSE objective does not introduce any conceptual or theoretical inconsistency with the identifiability assumptions required by the underlying SVAR model.

A.4 ADDITIONAL F1 RESULTS

Tables 6–9 present the F1 scores across all datasets under maximum lags of 3, 5, 10, and 15.

Table 6: F1 scores on all datasets with a maximum lag of 3

| Method | MoM 1 | MoM 2 | Ingestion | Web 1 | Web 2 | AntiV 1 | AntiV 2 | SWaT | Flood | Bavaria | East Germany |
|---|---|---|---|---|---|---|---|---|---|---|---|
| VAR | 0.1667 | 0.1622 | 0.2449 | 0.2222 | 0.2895 | 0.2047 | 0.2188 | 0.0590 | 0.0638 | 0.0064 | 0.0049 |
| MVGC | 0.0909 | 0.0000 | 0.0667 | 0.2059 | 0.2400 | 0.1613 | 0.1374 | 0.0758 | 0.0508 | TLE | TLE |
| cLSTM | 0.3390 | 0.3390 | 0.2500 | 0.2321 | 0.2500 | 0.1739 | 0.1739 | 0.0465 | 0.0465 | TLE | TLE |
| cMLP | 0.2857 | 0.1622 | 0.1379 | 0.1905 | 0.1081 | 0.2247 | 0.2857 | 0.0830 | 0.1139 | TLE | TLE |
| cRNN | 0.1714 | 0.1500 | 0.2069 | 0.1778 | 0.1667 | 0.2273 | 0.2903 | 0.0788 | 0.0991 | TLE | TLE |
| TCDF | 0.0000 | 0.0000 | 0.0000 | 0.0000 | 0.1053 | 0.1818 | 0.2000 | 0.0000 | 0.0000 | TLE | TLE |
| PCMCI+ | 0.0000 | 0.0000 | 0.0000 | 0.3243 | 0.1579 | 0.0357 | 0.0833 | TLE | TLE | TLE | TLE |
| tsFCI | 0.2286 | 0.0870 | 0.0000 | 0.1818 | 0.1714 | 0.1852 | 0.1159 | TLE | TLE | TLE | TLE |
| PCGCE | 0.0909 | 0.1538 | 0.1935 | 0.2143 | 0.1961 | 0.2250 | 0.2400 | 0.0820 | 0.0901 | TLE | TLE |
| DYNOTEARS | 0.2857 | 0.2353 | 0.1538 | 0.2623 | 0.2895 | 0.1905 | 0.2056 | 0.0440 | 0.1311 | TLE | TLE |
| VARLiNGAM | 0.0000 | 0.0000 | 0.3846 | 0.2593 | 0.2667 | 0.1923 | 0.2188 | 0.0677 | 0.0326 | TLE | TLE |
| TiMINo | 0.1538 | 0.1818 | 0.0000 | 0.0000 | 0.0000 | 0.0000 | 0.0000 | 0.0339 | 0.0000 | TLE | TLE |
| Our Method | 0.4000 | 0.3415 | 0.2769 | 0.3243 | 0.3243 | 0.1830 | 0.1875 | 0.2162 | 0.3125 | 0.1745 | 0.1212 |

Table 7: F1 scores on all datasets with a maximum lag of 5

| Method | MoM 1 | MoM 2 | Ingestion | Web 1 | Web 2 | AntiV 1 | AntiV 2 | SWaT | Flood | Bavaria | East Germany |
|---|---|---|---|---|---|---|---|---|---|---|---|
| VAR | 0.2632 | 0.2051 | 0.2400 | 0.2540 | 0.2857 | 0.2047 | 0.2188 | 0.0565 | 0.0607 | 0.0059 | 0.0045 |
| MVGC | 0.0833 | 0.1333 | 0.0667 | 0.2188 | 0.2368 | 0.1185 | 0.1212 | 0.0767 | 0.0559 | TLE | TLE |
| cLSTM | 0.3390 | 0.3390 | 0.2500 | 0.2321 | 0.2500 | 0.1739 | 0.1739 | 0.0279 | 0.0465 | TLE | TLE |
| cMLP | 0.2857 | 0.1622 | 0.1538 | 0.1860 | 0.1000 | 0.2247 | 0.2571 | 0.0902 | 0.1203 | TLE | TLE |
| cRNN | 0.1714 | 0.1905 | 0.2143 | 0.1818 | 0.1143 | 0.2353 | 0.3030 | 0.0788 | 0.0952 | TLE | TLE |
| TCDF | 0.0000 | 0.0000 | 0.0000 | 0.0000 | 0.1053 | 0.1818 | 0.2000 | 0.0000 | 0.0000 | TLE | TLE |
| PCMCI+ | 0.0000 | 0.1000 | 0.0000 | 0.2703 | 0.2051 | 0.0323 | 0.0800 | TLE | TLE | TLE | TLE |
| tsFCI | 0.2286 | 0.0870 | 0.0000 | 0.1875 | 0.1212 | 0.1250 | 0.1270 | TLE | TLE | TLE | TLE |
| PCGCE | 0.0833 | 0.0769 | 0.2286 | 0.1961 | 0.1132 | 0.2105 | 0.2632 | 0.0648 | 0.1116 | TLE | TLE |
| DYNOTEARS | 0.1250 | 0.3000 | 0.1538 | 0.2593 | 0.3462 | 0.1765 | 0.2131 | 0.0206 | 0.1342 | TLE | TLE |
| VARLiNGAM | 0.0000 | 0.0909 | 0.2500 | 0.2373 | 0.2535 | 0.1584 | 0.1655 | 0.0660 | 0.0403 | TLE | TLE |
| TiMINo | 0.0000 | 0.1667 | 0.0000 | 0.0000 | 0.0000 | 0.0000 | 0.0000 | 0.0580 | 0.0000 | TLE | TLE |
| Our Method | 0.4615 | 0.3784 | 0.2769 | 0.2697 | 0.2899 | 0.1916 | 0.2014 | 0.2202 | 0.3000 | 0.1751 | 0.1351 |

Table 8: F1 scores on all datasets with a maximum lag of 10

| Method | MoM 1 | MoM 2 | Ingestion | Web 1 | Web 2 | AntiV 1 | AntiV 2 | SWaT | Flood | Bavaria | East Germany |
|---|---|---|---|---|---|---|---|---|---|---|---|
| VAR | 0.2500 | 0.2927 | 0.2264 | 0.2647 | 0.2785 | 0.2016 | 0.2154 | 0.0539 | 0.0574 | 0.0056 | 0.0041 |
| MVGC | 0.0833 | 0.0000 | 0.0667 | 0.2258 | 0.2368 | 0.1194 | 0.1408 | 0.0787 | 0.0572 | TLE | TLE |
| cLSTM | 0.3390 | 0.3390 | 0.2500 | 0.2321 | 0.2500 | 0.1739 | 0.1739 | 0.0000 | 0.0466 | TLE | TLE |
| cMLP | 0.1622 | 0.1463 | 0.1481 | 0.1778 | 0.1053 | 0.2198 | 0.2353 | 0.0833 | 0.1158 | TLE | TLE |
| cRNN | 0.1667 | 0.1500 | 0.2069 | 0.1818 | 0.1111 | 0.2299 | 0.2985 | 0.0729 | 0.1009 | TLE | TLE |
| TCDF | 0.0000 | 0.0000 | 0.0000 | 0.0000 | 0.1053 | 0.1818 | 0.2000 | 0.0000 | 0.0000 | TLE | TLE |
| PCMCI+ | 0.0000 | 0.0000 | 0.0000 | 0.2927 | 0.2051 | 0.0597 | 0.1071 | TLE | TLE | TLE | TLE |
| tsFCI | 0.2353 | 0.0870 | 0.0000 | 0.2000 | 0.1176 | 0.0000 | 0.0000 | TLE | TLE | TLE | TLE |
| PCGCE | 0.0000 | 0.0714 | 0.2222 | 0.2917 | 0.1538 | 0.1772 | 0.2933 | 0.0827 | 0.0813 | TLE | TLE |
| DYNOTEARS | 0.2353 | 0.2222 | 0.1538 | 0.2759 | 0.3673 | 0.1757 | 0.2299 | 0.0215 | 0.1333 | TLE | TLE |
| VARLiNGAM | 0.2667 | 0.0000 | 0.2727 | 0.2545 | 0.2308 | 0.1856 | 0.1642 | 0.0621 | 0.0362 | TLE | TLE |
| TiMINo | 0.1429 | 0.2667 | 0.0000 | 0.0000 | 0.0000 | 0.0000 | 0.0000 | 0.0435 | 0.0000 | TLE | TLE |
| Our Method | 0.4286 | 0.4000 | 0.2812 | 0.2697 | 0.3014 | 0.1905 | 0.2162 | 0.2182 | 0.3226 | 0.1860 | 0.1388 |

Table 9: F1 scores on all datasets with a maximum lag of 15.

| Method | MoM 1 | MoM 2 | Ingestion | Web 1 | Web 2 | AntiV 1 | AntiV 2 | SWaT | Flood | Bavaria | East Germany |
|---|---|---|---|---|---|---|---|---|---|---|---|
| VAR | 0.2927 | 0.2500 | 0.2308 | 0.2535 | 0.2683 | 0.2000 | 0.2137 | 0.0512 | 0.0561 | 0.0054 | 0.0039 |
| MVGC | 0.0833 | 0.0000 | 0.0667 | 0.2258 | 0.2192 | 0.1231 | 0.1630 | 0.0877 | 0.0556 | TLE | TLE |
| cLSTM | 0.3390 | 0.3390 | 0.2500 | 0.2321 | 0.2500 | 0.1739 | 0.1739 | 0.0000 | 0.0466 | TLE | TLE |
| cMLP | 0.2162 | 0.1579 | 0.1481 | 0.1905 | 0.1053 | 0.2247 | 0.2647 | 0.0844 | 0.1172 | TLE | TLE |
| cRNN | 0.2222 | 0.1463 | 0.2069 | 0.1905 | 0.1111 | 0.2247 | 0.2941 | 0.0753 | 0.1026 | TLE | TLE |
| TCDF | 0.0000 | 0.0000 | 0.0000 | 0.0000 | 0.1053 | 0.1818 | 0.2000 | 0.0000 | 0.0000 | TLE | TLE |
| PCMCI+ | 0.2727 | 0.0000 | 0.0000 | 0.3000 | 0.2051 | 0.0870 | 0.0984 | TLE | TLE | TLE | TLE |
| tsFCI | 0.2353 | 0.0870 | 0.0000 | 0.2000 | 0.1176 | 0.0000 | 0.0000 | TLE | TLE | TLE | TLE |
| PCGCE | 0.0000 | 0.0714 | 0.1579 | 0.2353 | 0.1852 | 0.1707 | 0.2338 | 0.0722 | 0.0735 | TLE | TLE |
| DYNOTEARS | 0.3636 | 0.2500 | 0.1538 | 0.2963 | 0.3600 | 0.1892 | 0.2273 | 0.0230 | 0.1560 | TLE | TLE |
| VARLiNGAM | 0.0000 | 0.1333 | 0.2857 | 0.2642 | 0.2368 | 0.1818 | 0.1395 | 0.0618 | 0.0407 | TLE | TLE |
| TiMINo | 0.1538 | 0.1667 | 0.1667 | 0.0000 | 0.0000 | 0.0000 | 0.0000 | 0.0364 | 0.0000 | TLE | TLE |
| Our Method | 0.3415 | 0.4091 | 0.2769 | 0.2697 | 0.3143 | 0.1893 | 0.2222 | 0.1818 | 0.3438 | 0.1997 | 0.1279 |

First, our method consistently achieves top or near-top performance across datasets and lag settings. For example, it outperforms all baselines on MoM1/2, Ingestion, and Web datasets under most lags, and maintains competitive accuracy on the AntiVirus datasets, where most methods struggle due to their sparse and irregular event-driven nature.

Second, baseline methods often show dataset-specific strengths but lack robustness. VARLiNGAM performs well on Ingestion but poorly on MoM datasets, DYNOTEARS excels on Web1/Web2 but degrades on MoM and AntiVirus, while cRNN and PCGCE perform relatively better on AntiVirus datasets. This indicates that these methods may overfit to particular data characteristics rather than generalize across domains.

Third, increasing the maximum lag generally benefits our method, especially on larger datasets such as Flood, Bavaria, and East Germany, where it achieves the strongest performance once longer temporal dependencies are captured. In contrast, many baselines either fail to scale to these datasets (TLE) or show little improvement when lag increases.

Overall, these results confirm that our method provides both stable and scalable causal discovery performance across heterogeneous datasets, whereas existing baselines are less consistent and often limited to narrow data regimes.

## A.5 ANALYSIS OF ANTIVIRUS DATASET PERFORMANCE AND THE ROLE OF PRUNING THRESHOLDS

The AntiV1 and AntiV2 datasets constitute some of the most challenging cases in our benchmark. As discussed by Aït-Bachir et al. (2023), these datasets contain sparse, irregular, and event-driven temporal dynamics arising from mixed sampling rates, partial sleeping behaviour, and interpolation steps. Under such circumstances, small but meaningful causal effects can be difficult to distinguish from noise, while bursty fluctuations may be retained as false positives.

In our main experiments, we used a fixed pruning threshold of 0.01 across all datasets. While this value works well for most settings, it can be suboptimal for AntiV1 and AntiV2 due to their irregular dynamics. Table 10 shows the F1 scores obtained when varying the pruning threshold across different lags. The results reveal that the weaker performance reported in the main text is largely attributable to the fixed pruning threshold rather than the underlying modelling capacity of our method.

For both AntiV1 and AntiV2, increasing the pruning threshold generally leads to substantial improvements in F1 score. For instance, on AntiV1 at lag 5, the F1 score rises from 0.1916 (threshold 0.01) to 0.2667 at threshold 0.15. A similar trend is observed for AntiV2, where the F1 score at lag 3 increases from 0.1875 (threshold 0.01) to 0.3030 at threshold 0.15. These results demonstrate that, with an appropriate pruning threshold, our method achieves markedly stronger performance on the AntiVirus datasets.

This analysis suggests that the AntiV performance reported in the main experiments reflects the sensitivity of pruning to irregular time-series characteristics rather than a fundamental limitation of the model. The improvement obtained through threshold tuning highlights the importance of dataset-dependent sparsity control and motivates future work toward adaptive or stability-based pruning strategies that can automatically adjust to heterogeneous temporal behaviours.

Table 10: F1 scores on AntiV1 and AntiV2 datasets with different pruning thresholds across lags. A fixed pruning threshold of 0.01 is used in our main experiments.

| Threshold | AntiV1 | | | | AntiV2 | | | |
|---|---|---|---|---|---|---|---|---|
| | 0.01 | 0.05 | 0.10 | 0.15 | 0.01 | 0.05 | 0.10 | 0.15 |
| Lag 3 | 0.1830 | 0.2128 | 0.2034 | 0.2222 | 0.1875 | 0.2273 | 0.2273 | 0.3030 |
| Lag 5 | 0.1916 | 0.2500 | 0.2500 | 0.2667 | 0.2014 | 0.2857 | 0.2857 | 0.2581 |
| Lag 10 | 0.1905 | 0.2059 | 0.2400 | 0.2400 | 0.2162 | 0.2581 | 0.2581 | 0.2581 |
| Lag 15 | 0.1893 | 0.1778 | 0.1778 | 0.1778 | 0.2222 | 0.2500 | 0.2500 | 0.2500 |

## A.6 ADDITIONAL RECALL RESULTS

Tables 11–14 summarize recall across all datasets and lag settings. Several consistent patterns emerge.

Table 11: Recall on all datasets with a maximum lag of 3. *TLE*: time limit exceeded (3 hours).

| Method | MoM 1 | MoM 2 | Ingestion | Web 1 | Web 2 | AntiV 1 | AntiV 2 | SWaT | Flood | Bavaria | East Germany |
|---|---|---|---|---|---|---|---|---|---|---|---|
| VAR | 0.3000 | 0.3000 | 0.6667 | 0.5000 | 0.7857 | 0.8125 | 0.8750 | 0.7297 | 1.0000 | 0.9816 | 0.9339 |
| MVGC | 0.1000 | 0.0000 | 0.1111 | 0.5000 | 0.6429 | 0.6250 | 0.5625 | 0.7297 | 0.5238 | TLE | TLE |
| cLSTM | 1.0000 | 1.0000 | 1.0000 | 0.9286 | 1.0000 | 1.0000 | 1.0000 | 1.0000 | 0.9762 | TLE | TLE |
| cMLP | 0.5000 | 0.3000 | 0.2222 | 0.2857 | 0.1429 | 0.6250 | 0.6250 | 0.2703 | 0.3810 | TLE | TLE |
| cRNN | 0.3000 | 0.3000 | 0.3333 | 0.2857 | 0.2143 | 0.6250 | 0.5625 | 0.3636 | 0.4048 | TLE | TLE |
| TCDF | 0.0000 | 0.0000 | 0.0000 | 0.0000 | 0.0714 | 0.1250 | 0.1250 | 0.0000 | 0.0000 | TLE | TLE |
| PCMCI+ | 0.0000 | 0.0000 | 0.0000 | 0.4286 | 0.2143 | 0.0625 | 0.1250 | TLE | TLE | TLE | TLE |
| tsFCI | 0.4000 | 0.1000 | 0.0000 | 0.2143 | 0.2143 | 0.3125 | 0.2500 | TLE | TLE | TLE | TLE |
| PCGCE | 0.1000 | 0.2000 | 0.3333 | 0.4286 | 0.3571 | 0.5625 | 0.5625 | 0.2703 | 0.2381 | TLE | TLE |
| DYNOTEARS | 0.3000 | 0.2000 | 0.1111 | 0.5714 | 0.6429 | 0.8750 | 0.6875 | 0.0541 | 0.2857 | TLE | TLE |
| VARLiNGAM | 0.0000 | 0.0000 | 0.5556 | 0.5000 | 0.5714 | 0.6250 | 0.8750 | 0.7297 | 0.3333 | TLE | TLE |
| TiMINo | 0.1000 | 0.1000 | 0.0000 | 0.0000 | 0.0000 | 0.0000 | 0.0000 | 0.0270 | 0.0000 | TLE | TLE |
| Our method | 0.8000 | 0.7000 | 1.0000 | 0.8571 | 0.8571 | 0.8750 | 0.7500 | 0.3243 | 0.2381 | 0.1061 | 0.0676 |

First, our method achieves high recall on small- and medium-scale datasets such as MoM1/2, Ingestion, and Web, where it often reaches values close to or equal to 1.0. This indicates that our

Table 12: Recall on all datasets with a maximum lag of 5. *TLE*: time limit exceeded (3 hours).

| Method | MoM 1 | MoM 2 | Ingestion | Web 1 | Web 2 | AntiV 1 | AntiV 2 | SWaT | Flood | Bavaria | East Germany |
|---|---|---|---|---|---|---|---|---|---|---|---|
| VAR | 0.5000 | 0.4000 | 0.6667 | 0.5714 | 0.7857 | 0.8125 | 0.8750 | 0.7297 | 1.0000 | 0.9837 | 0.9386 |
| MVGC | 0.1000 | 0.2000 | 0.1111 | 0.5000 | 0.6429 | 0.5000 | 0.5000 | 0.7027 | 0.6429 | TLE | TLE |
| TCDF | 0.0000 | 0.0000 | 0.0000 | 0.0000 | 0.0714 | 0.1250 | 0.1250 | 1.0000 | 0.9762 | TLE | TLE |
| PCMCI+ | 0.0000 | 0.1000 | 0.0000 | 0.3571 | 0.2857 | 0.0625 | 0.1250 | 0.2973 | 0.3810 | TLE | TLE |
| tsFCI | 0.4000 | 0.1000 | 0.0000 | 0.2143 | 0.1429 | 0.1875 | 0.2500 | 0.3636 | 0.3810 | TLE | TLE |
| PCGCE | 0.1000 | 0.1000 | 0.4444 | 0.3571 | 0.2143 | 0.5000 | 0.6250 | 0.0000 | 0.0000 | TLE | TLE |
| cLSTM | 1.0000 | 1.0000 | 1.0000 | 0.9286 | 1.0000 | 1.0000 | 1.0000 | TLE | TLE | TLE | TLE |
| cMLP | 0.5000 | 0.3000 | 0.2222 | 0.2857 | 0.1429 | 0.6250 | 0.5625 | TLE | TLE | TLE | TLE |
| cRNN | 0.3000 | 0.4000 | 0.3333 | 0.2857 | 0.1429 | 0.6250 | 0.6250 | 0.2162 | 0.3095 | TLE | TLE |
| DYNOTEARS | 0.1000 | 0.3000 | 0.1111 | 0.5000 | 0.6429 | 0.7500 | 0.8125 | 0.0270 | 0.2381 | TLE | TLE |
| VARLiNGAM | 0.0000 | 0.1000 | 0.3333 | 0.5000 | 0.6429 | 0.5000 | 0.7500 | 0.7838 | 0.3333 | TLE | TLE |
| TiMINo | 0.0000 | 0.1000 | 0.0000 | 0.0000 | 0.0000 | 0.0000 | 0.0000 | 0.0541 | 0.0000 | TLE | TLE |
| Our method | 0.9000 | 0.7000 | 1.0000 | 0.8571 | 0.7143 | 1.0000 | 0.8750 | 0.3243 | 0.2143 | 0.1061 | 0.0768 |

Table 13: Recall on all datasets with a maximum lag of 10. *TLE*: time limit exceeded (3 hours).

| Method | MoM 1 | MoM 2 | Ingestion | Web 1 | Web 2 | AntiV 1 | AntiV 2 | SWaT | Flood | Bavaria | East Germany |
|---|---|---|---|---|---|---|---|---|---|---|---|
| VAR | 0.5000 | 0.6000 | 0.6667 | 0.6429 | 0.7857 | 0.8125 | 0.8750 | 0.7568 | 1.0000 | 0.9898 | 0.9493 |
| MVGC | 0.1000 | 0.0000 | 0.1111 | 0.5000 | 0.6429 | 0.5000 | 0.6250 | 0.6757 | 0.7381 | TLE | TLE |
| cLSTM | 1.0000 | 1.0000 | 1.0000 | 0.9286 | 1.0000 | 1.0000 | 1.0000 | 0.0000 | 0.9762 | TLE | TLE |
| cMLP | 0.3000 | 0.3000 | 0.2222 | 0.2857 | 0.1429 | 0.6250 | 0.5000 | 0.2703 | 0.3571 | TLE | TLE |
| cRNN | 0.3000 | 0.3000 | 0.3333 | 0.2857 | 0.1429 | 0.6250 | 0.6250 | 0.3182 | 0.3810 | TLE | TLE |
| TCDF | 0.0000 | 0.0000 | 0.0000 | 0.0000 | 0.0714 | 0.1250 | 0.1250 | 0.0000 | 0.0000 | TLE | TLE |
| PCMICI+ | 0.0000 | 0.0000 | 0.0000 | 0.4286 | 0.2857 | 0.1250 | 0.1875 | TLE | TLE | TLE | TLE |
| tsFCI | 0.4000 | 0.1000 | 0.0000 | 0.2143 | 0.1429 | 0.0000 | 0.0000 | TLE | TLE | TLE | TLE |
| PCGCE | 0.0000 | 0.1000 | 0.4444 | 0.5000 | 0.2857 | 0.4375 | 0.6875 | 0.2973 | 0.2381 | TLE | TLE |
| DYNOTEARS | 0.2000 | 0.2000 | 0.1111 | 0.5714 | 0.6429 | 0.8125 | 0.6250 | 0.0270 | 0.2143 | TLE | TLE |
| VARLiNGAM | 0.2000 | 0.0000 | 0.3333 | 0.5000 | 0.6429 | 0.5625 | 0.6875 | 0.7838 | 0.2381 | TLE | TLE |
| TiMINo | 0.1000 | 0.2000 | 0.0000 | 0.0000 | 0.0000 | 0.0000 | 0.0000 | 0.0270 | 0.0000 | TLE | TLE |
| Our method | 0.9000 | 0.8000 | 1.0000 | 0.8571 | 0.7857 | 1.0000 | 1.0000 | 0.3243 | 0.2381 | 0.1163 | 0.0783 |

Table 14: Recall on all datasets with a maximum lag of 15. *TLE*: time limit exceeded (3 hours) limit.

| Method | MoM 1 | MoM 2 | Ingestion | Web 1 | Web 2 | AntiV 1 | AntiV 2 | SWaT | Flood | Bavaria | East Germany |
|---|---|---|---|---|---|---|---|---|---|---|---|
| VAR | 0.6000 | 0.5000 | 0.6667 | 0.6429 | 0.7857 | 0.8125 | 0.8750 | 0.7297 | 1.0000 | 0.9898 | 0.9508 |
| MVGC | 0.1000 | 0.0000 | 0.1111 | 0.5000 | 0.5714 | 0.5000 | 0.6875 | 0.7027 | 0.7619 | TLE | TLE |
| cLSTM | 1.0000 | 1.0000 | 1.0000 | 0.9286 | 1.0000 | 1.0000 | 1.0000 | 0.0000 | 0.9762 | TLE | TLE |
| cMLP | 0.4000 | 0.3000 | 0.2222 | 0.2857 | 0.1429 | 0.6250 | 0.5625 | 0.2703 | 0.3571 | TLE | TLE |
| cRNN | 0.4000 | 0.3000 | 0.3333 | 0.2857 | 0.1429 | 0.6250 | 0.6250 | 0.3182 | 0.3810 | TLE | TLE |
| TCDF | 0.0000 | 0.0000 | 0.0000 | 0.0000 | 0.0714 | 0.1250 | 0.1250 | 0.0000 | 0.0000 | TLE | TLE |
| PCMCI+ | 0.3000 | 0.0000 | 0.0000 | 0.4286 | 0.2857 | 0.1875 | 0.1875 | TLE | TLE | TLE | TLE |
| tsFCI | 0.4000 | 0.1000 | 0.0000 | 0.2143 | 0.1429 | 0.0000 | 0.0000 | TLE | TLE | TLE | TLE |
| PCGCE | 0.0000 | 0.1000 | 0.3333 | 0.4286 | 0.3571 | 0.4375 | 0.5625 | 0.2703 | 0.2143 | TLE | TLE |
| DYNOTEARS | 0.4000 | 0.2000 | 0.1111 | 0.5714 | 0.6429 | 0.8750 | 0.6250 | 0.0270 | 0.2619 | TLE | TLE |
| VARLiNGAM | 0.0000 | 0.1000 | 0.3333 | 0.5000 | 0.6429 | 0.5000 | 0.5625 | 0.7838 | 0.2857 | TLE | TLE |
| TiMINo | 0.1000 | 0.1000 | 0.1111 | 0.0000 | 0.0000 | 0.0000 | 0.0000 | 0.0270 | 0.0000 | TLE | TLE |
| Our method | 0.7000 | 0.9000 | 1.0000 | 0.8571 | 0.7857 | 1.0000 | 1.0000 | 0.2703 | 0.2619 | 0.1286 | 0.0722 |

approach is effective at recovering true causal edges in settings with limited dimensionality, and performs comparably or better than strong baselines such as cLSTM.

Second, on the AntiVirus datasets, our method maintains strong recall (0.75–1.0), highlighting its ability to capture the majority of ground-truth causal relations even in event-driven time series. By contrast, many baselines either underperform or show large variance across different lags.

Third, recall drops substantially on large-scale datasets such as Flood, Bavaria, and East Germany, where our method records much lower values (e.g., below 0.3 in several cases). In contrast, VAR consistently achieves nearly perfect recall on these datasets, though at the cost of very low precision (as shown in Section A.7). This suggests that our pruning strategy is conservative, favoring precision over recall in high-dimensional settings.

Overall, these results show that our method provides balanced recall in small- and medium-scale settings while remaining competitive on more challenging datasets. However, the decline in recall on large-scale datasets underscores the limitation of using a fixed pruning threshold, which may discard weak but meaningful causal effects. Developing adaptive pruning strategies could help alleviate this trade-off and improve recall without sacrificing precision.

## A.7 Additional Precision Results

Tables 15–18 report precision across all datasets and lag settings. Several patterns can be observed.

Table 15: Precision on all datasets with a maximum lag of 3. *TLE*: time limit exceeded (3 hours).

| Method | MoM 1 | MoM 2 | Ingestion | Web 1 | Web 2 | AntiV 1 | AntiV 2 | SWaT | Flood | Bavaria | East Germany |
|---|---|---|---|---|---|---|---|---|---|---|---|
| VAR | 0.1154 | 0.1111 | 0.1500 | 0.1429 | 0.1774 | 0.1171 | 0.1250 | 0.0308 | 0.0329 | 0.0032 | 0.0025 |
| MVGC | 0.0833 | 0.0000 | 0.0476 | 0.1296 | 0.1475 | 0.0926 | 0.0783 | 0.0400 | 0.0267 | TLE | TLE |
| cLSTM | 0.2041 | 0.2041 | 0.1429 | 0.1327 | 0.1429 | 0.0952 | 0.0952 | 0.0238 | 0.0238 | TLE | TLE |
| cMLP | 0.2000 | 0.1111 | 0.1000 | 0.1429 | 0.0870 | 0.1370 | 0.1852 | 0.0490 | 0.0669 | TLE | TLE |
| cRNN | 0.1200 | 0.1000 | 0.1500 | 0.1290 | 0.1364 | 0.1389 | 0.1957 | 0.0442 | 0.0565 | TLE | TLE |
| TCDF | 0.0000 | 0.0000 | 0.0000 | 0.0000 | 0.2000 | 0.3333 | 0.5000 | 0.0000 | 0.0000 | TLE | TLE |
| PCMCI+ | 0.0000 | 0.0000 | 0.0000 | 0.2609 | 0.1250 | 0.0250 | 0.0625 | TLE | TLE | TLE | TLE |
| tsFCI | 0.1600 | 0.0769 | 0.0000 | 0.1579 | 0.1429 | 0.1316 | 0.0755 | TLE | TLE | TLE | TLE |
| PCGCE | 0.0833 | 0.1250 | 0.1364 | 0.1429 | 0.1351 | 0.1406 | 0.1525 | 0.0483 | 0.0556 | TLE | TLE |
| DYNOTEARS | 0.2727 | 0.2857 | 0.2500 | 0.1702 | 0.2500 | 0.1069 | 0.1209 | 0.0370 | 0.0851 | TLE | TLE |
| VARLiNGAM | 0.0000 | 0.0000 | 0.2941 | 0.1750 | 0.1739 | 0.1136 | 0.1250 | 0.0355 | 0.0171 | TLE | TLE |
| TiMINo | 0.3333 | 1.0000 | 0.0000 | 0.0000 | 0.0000 | 0.0000 | 0.0000 | 0.0455 | 0.0000 | TLE | TLE |
| Our method | 0.2667 | 0.2258 | 0.1607 | 0.2000 | 0.2000 | 0.1022 | 0.1071 | 0.1622 | 0.4545 | 0.4906 | 0.5867 |

Table 16: Precision on all datasets with a maximum lag of 5. *TLE*: time limit exceeded (3 hours).

| Method | MoM 1 | MoM 2 | Ingestion | Web 1 | Web 2 | AntiV 1 | AntiV 2 | SWaT | Flood | Bavaria | East Germany |
|---|---|---|---|---|---|---|---|---|---|---|---|
| VAR | 0.1786 | 0.1379 | 0.1463 | 0.1633 | 0.1746 | 0.1171 | 0.1250 | 0.0294 | 0.0313 | 0.0030 | 0.0022 |
| MVGC | 0.0714 | 0.1000 | 0.0476 | 0.1400 | 0.1452 | 0.0672 | 0.0690 | 0.0406 | 0.0292 | TLE | TLE |
| cLSTM | 0.2041 | 0.2041 | 0.1429 | 0.1327 | 0.1429 | 0.0952 | 0.0952 | 0.0141 | 0.0238 | TLE | TLE |
| cMLP | 0.2000 | 0.1111 | 0.1176 | 0.1379 | 0.0769 | 0.1370 | 0.1667 | 0.0531 | 0.0714 | TLE | TLE |
| cRNN | 0.1200 | 0.1250 | 0.1579 | 0.1333 | 0.0952 | 0.1449 | 0.2000 | 0.0442 | 0.0544 | TLE | TLE |
| TCDF | 0.0000 | 0.0000 | 0.0000 | 0.0000 | 0.2000 | 0.3333 | 0.5000 | 0.0000 | 0.0000 | TLE | TLE |
| PCMCI+ | 0.0000 | 0.1000 | 0.0000 | 0.2174 | 0.1600 | 0.0217 | 0.0588 | TLE | TLE | TLE | TLE |
| tsFCI | 0.1600 | 0.0769 | 0.0000 | 0.1667 | 0.1053 | 0.0938 | 0.0851 | TLE | TLE | TLE | TLE |
| PCGCE | 0.0714 | 0.0625 | 0.1538 | 0.1351 | 0.0769 | 0.1333 | 0.1667 | 0.0381 | 0.0681 | TLE | TLE |
| DYNOTEARS | 0.1667 | 0.3000 | 0.2500 | 0.1750 | 0.2368 | 0.1000 | 0.1226 | 0.0167 | 0.0935 | TLE | TLE |
| VARLiNGAM | 0.0000 | 0.0833 | 0.2000 | 0.1556 | 0.1579 | 0.0941 | 0.0930 | 0.0344 | 0.0215 | TLE | TLE |
| TiMINo | 0.0000 | 0.5000 | 0.0000 | 0.0000 | 0.0000 | 0.0000 | 0.0000 | 0.0625 | 0.0000 | TLE | TLE |
| Our method | 0.3103 | 0.2593 | 0.1607 | 0.1600 | 0.1818 | 0.1060 | 0.1138 | 0.1667 | 0.5000 | 0.5000 | 0.5618 |

Table 17: Precision on all datasets with a maximum lag of 10. *TLE*: time limit exceeded (3 hours).

| Method | MoM 1 | MoM 2 | Ingestion | Web 1 | Web 2 | AntiV 1 | AntiV 2 | SWaT | Flood | Bavaria | East Germany |
|---|---|---|---|---|---|---|---|---|---|---|---|
| VAR | 0.1667 | 0.1935 | 0.1364 | 0.1667 | 0.1692 | 0.1150 | 0.1228 | 0.0280 | 0.0296 | 0.0028 | 0.0021 |
| MVGC | 0.0714 | 0.0000 | 0.0476 | 0.1458 | 0.1452 | 0.0678 | 0.0794 | 0.0418 | 0.0298 | TLE | TLE |
| cLSTM | 0.2041 | 0.2041 | 0.1429 | 0.1327 | 0.1429 | 0.0952 | 0.0952 | 0.0000 | 0.0239 | TLE | TLE |
| cMLP | 0.1111 | 0.0968 | 0.1111 | 0.1290 | 0.0833 | 0.1333 | 0.1538 | 0.0493 | 0.0691 | TLE | TLE |
| cRNN | 0.1154 | 0.1000 | 0.1500 | 0.1333 | 0.0909 | 0.1408 | 0.1961 | 0.0412 | 0.0582 | TLE | TLE |
| TCDF | 0.0000 | 0.0000 | 0.0000 | 0.0000 | 0.2000 | 0.3333 | 0.5000 | 0.0000 | 0.0000 | TLE | TLE |
| PCMICI+ | 0.0000 | 0.0000 | 0.0000 | 0.2222 | 0.1600 | 0.0392 | 0.0750 | TLE | TLE | TLE | TLE |
| tsFCI | 0.1667 | 0.0769 | 0.0000 | 0.1875 | 0.1000 | 0.0000 | 0.0000 | TLE | TLE | TLE | TLE |
| PCGCE | 0.0000 | 0.0556 | 0.1481 | 0.2059 | 0.1053 | 0.1111 | 0.1864 | 0.0480 | 0.0490 | TLE | TLE |
| DYNOTEARS | 0.2857 | 0.2500 | 0.2500 | 0.1818 | 0.2571 | 0.0985 | 0.1408 | 0.0179 | 0.0968 | TLE | TLE |
| VARLiNGAM | 0.4000 | 0.0000 | 0.2308 | 0.1707 | 0.1406 | 0.1111 | 0.0932 | 0.0323 | 0.0196 | TLE | TLE |
| TiMINo | 0.2500 | 0.4000 | 0.0000 | 0.0000 | 0.0000 | 0.0000 | 0.0000 | 0.1111 | 0.0000 | TLE | TLE |
| Our method | 0.2812 | 0.2667 | 0.1636 | 0.1600 | 0.1864 | 0.1053 | 0.1212 | 0.1644 | 0.5000 | 0.4634 | 0.6071 |

Table 18: Precision on all datasets with a maximum lag of 15. *TLE*: time limit exceeded (3 hours).

| Method | MoM 1 | MoM 2 | Ingestion | Web 1 | Web 2 | AntiV 1 | AntiV 2 | SWaT | Flood | Bavaria | East Germany |
|---|---|---|---|---|---|---|---|---|---|---|---|
| VAR | 0.1935 | 0.1667 | 0.1395 | 0.1579 | 0.1618 | 0.1140 | 0.1217 | 0.0265 | 0.0289 | 0.0027 | 0.0020 |
| MVGC | 0.0714 | 0.0000 | 0.0476 | 0.1458 | 0.1356 | 0.0702 | 0.0924 | 0.0468 | 0.0288 | TLE | TLE |
| cLSTM | 0.2041 | 0.2041 | 0.1429 | 0.1327 | 0.1429 | 0.0952 | 0.0952 | 0.0000 | 0.0239 | TLE | TLE |
| cMLP | 0.1481 | 0.1071 | 0.1111 | 0.1429 | 0.0833 | 0.1370 | 0.1731 | 0.0500 | 0.0701 | TLE | TLE |
| cRNN | 0.1538 | 0.0968 | 0.1500 | 0.1429 | 0.0909 | 0.1370 | 0.1923 | 0.0427 | 0.0593 | TLE | TLE |
| TCDF | 0.0000 | 0.0000 | 0.0000 | 0.0000 | 0.2000 | 0.3333 | 0.5000 | 0.0000 | 0.0000 | TLE | TLE |
| PCMCI+ | 0.2500 | 0.0000 | 0.0000 | 0.2308 | 0.1600 | 0.0566 | 0.0667 | TLE | TLE | TLE | TLE |
| tsFCI | 0.1667 | 0.0769 | 0.0000 | 0.1875 | 0.1000 | 0.0000 | 0.0000 | TLE | TLE | TLE | TLE |
| PCGCE | 0.0000 | 0.0556 | 0.1034 | 0.1622 | 0.1250 | 0.1061 | 0.1475 | 0.0417 | 0.0443 | TLE | TLE |
| DYNOTEARS | 0.3333 | 0.3333 | 0.2500 | 0.2000 | 0.2500 | 0.1061 | 0.1389 | 0.0200 | 0.1111 | TLE | TLE |
| VARLiNGAM | 0.0000 | 0.2000 | 0.2500 | 0.1795 | 0.1452 | 0.1111 | 0.0796 | 0.0322 | 0.0219 | TLE | TLE |
| TiMINO | 0.3333 | 0.5000 | 0.3333 | 0.0000 | 0.0000 | 0.0000 | 0.0000 | 0.0556 | 0.0000 | TLE | TLE |
| Our method | 0.2258 | 0.2647 | 0.1607 | 0.1600 | 0.1964 | 0.1046 | 0.1250 | 0.1370 | 0.5000 | 0.4468 | 0.5595 |

First, our method consistently achieves the highest precision on large-scale datasets such as Flood, Bavaria, and East Germany, where it maintains values above 0.44 and up to 0.61. This demonstrates that our approach is effective at avoiding false positives when scaling to high-dimensional settings, in contrast to most baselines which either fail to scale (TLE) or degrade sharply in precision.

Second, on small- and medium-scale datasets, our method shows more moderate precision compared to some baselines. For example, DYNOTEARS and VARLiNGAM occasionally achieve higher precision on Web or Ingestion datasets, while TiMINo can reach very high precision in isolated cases (e.g., MoM2 with lag 3). However, these methods tend to be unstable across datasets, often collapsing to zero or near-zero precision in other settings, whereas our method remains consistently competitive.

Third, precision on the AntiVirus datasets remains relatively low for all methods, including ours, reflecting the difficulty of handling sparse and irregular event-driven signals. This again highlights the sensitivity of pruning strategies in such settings, as discussed in Section A.5.

Overall, these results confirm that while some baselines may excel on particular datasets, our method provides the best trade-off between stability and scalability. In particular, its strong precision on large-scale datasets shows its practical value for high-dimensional causal discovery tasks.

## A.8 ADDITIONAL CPU RUNTIME RESULTS

Tables 19–21 report the CPU runtime across all datasets and lag settings. On small-scale datasets (the first seven columns), our method is not the fastest—linear approaches such as VAR and PCGCE consistently achieve lower runtime. However, our method still completes within seconds to tens of seconds, which remains practical for these scales.

Table 19: Runtime (in seconds) for All Datasets with Lag = 3. TLE: time limit exceeded (3 hours).

| Method | MoM 1 | MoM 2 | Storm | Web 1 | Web 2 | AntiV 1 | AntiV 2 | SWaT | Flood | Bavaria | Germany |
|---|---|---|---|---|---|---|---|---|---|---|---|
| **VAR** | 0.000 | 0.000 | 0.000 | 0.015 | 0.016 | 0.006 | 0.002 | 0.09 | 0.05 | 5.32 | 9.49 |
| **MVGC** | 0.090 | 0.100 | 0.079 | 0.367 | 0.312 | 0.592 | 1.090 | 135.58 | 129.37 | TLE | TLE |
| **cLSTM** | 22.89 | 25.684 | 34.635 | 67.289 | 65.577 | 52.093 | 44.384 | 1262.41 | 227.32 | TLE | TLE |
| **cMLP** | 5.586 | 5.641 | 7.257 | 29.618 | 30.507 | 13.395 | 13.082 | 209.29 | 135.37 | TLE | TLE |
| **cRNN** | 17.165 | 17.413 | 21.860 | 62.714 | 62.004 | 47.574 | 47.899 | 1144.85 | 160.22 | TLE | TLE |
| **TCDF** | 8.69 | 9.02 | 9.94 | 58.36 | 58.55 | 16.65 | 17.81 | 550.15 | 258.03 | TLE | TLE |
| **PCMCI+** | 0.250 | 0.293 | 3.233 | 40.194 | 31.030 | 23.617 | 38.671 | TLE | TLE | TLE | TLE |
| **tsFCI** | 26.378 | 14.421 | 1.048 | 100.715 | 332.605 | 245.800 | 620.111 | TLE | TLE | TLE | TLE |
| **PCGCE** | 0.139 | 0.433 | 0.858 | 20.652 | 4.460 | 18.210 | 7.867 | 777.46 | 75.00 | TLE | TLE |
| **DYNOTEARS** | 2.194 | 1.701 | 0.031 | 15.649 | 37.256 | 10.595 | 161.464 | 740.44 | 259.33 | TLE | TLE |
| **VARLINGAM** | 0.088 | 0.122 | 0.128 | 2.189 | 2.201 | 0.437 | 0.750 | 409.37 | 15.02 | TLE | TLE |
| **TiMINo** | 1.760 | 1.809 | 2.759 | 17.630 | 21.774 | 6.888 | 6.539 | 1654.83 | 79.98 | TLE | TLE |
| **Our Method** | 9.679 | 8.174 | 8.180 | 11.881 | 14.797 | 8.146 | 10.065 | 23.52 | 23.29 | 716.42 | 1213.15 |

Table 20: Runtime (in seconds) for All Datasets with Lag = 5. TLE: time limit exceeded (3 hours).

| Method | MoM 1 | MoM 2 | Storm | Web 1 | Web 2 | AntiV 1 | AntiV 2 | SWaT | Flood | Bavaria | Germany |
|---|---|---|---|---|---|---|---|---|---|---|---|
| **VAR** | 0 | 0.00 | 0.00 | 0.02 | 0.02 | 0.02 | 0.02 | 0.17 | 0.10 | 12.08 | 19.69 |
| **MVGC** | 0.110 | 0.17 | 0.11 | 1.41 | 1.52 | 4.07 | 4.10 | 1064.92 | 816.72 | TLE | TLE |
| **cLSTM** | 20.117 | 24.24 | 41.86 | 94.96 | 92.77 | 72.17 | 73.23 | 4346.78 | 367.78 | TLE | TLE |
| **cMLP** | 6.736 | 6.91 | 9.08 | 30.46 | 36.89 | 18.81 | 21.35 | 261.13 | 138.31 | TLE | TLE |
| **cRNN** | 16.55 | 19.89 | 27.82 | 83.52 | 73.93 | 49.95 | 50.87 | 1845.22 | 274.41 | TLE | TLE |
| **TCDF** | 8.69 | 9.02 | 9.94 | 58.36 | 58.55 | 16.65 | 17.81 | 550.15 | 258.03 | TLE | TLE |
| **PCMCI+** | 0.190 | 0.37 | 6.32 | 187.47 | 78.93 | 87.51 | 65.56 | TLE | TLE | TLE | TLE |
| **tsFCI** | 29.77 | 19.36 | 0.94 | 131.07 | 203.35 | 368.48 | 786.60 | TLE | TLE | TLE | TLE |
| **PCGCE** | 0.180 | 0.54 | 0.91 | 13.65 | 4.65 | 22.43 | 9.72 | 871.41 | 75.22 | TLE | TLE |
| **DYNOTEARS** | 10.77 | 2.90 | 0.06 | 56.42 | 57.53 | 15.33 | 364.57 | 1897.80 | 313.75 | TLE | TLE |
| **VARLINGAM** | 0.093 | 0.13 | 0.21 | 4.75 | 4.58 | 1.06 | 1.07 | 1208.18 | 29.36 | TLE | TLE |
| **TiMINo** | 1.312 | 1.76 | 2.12 | 13.83 | 17.30 | 7.07 | 7.09 | 1051.04 | 79.85 | TLE | TLE |
| **Our Method** | 8.317 | 7.47 | 8.34 | 12.75 | 15.87 | 6.55 | 8.07 | 25.06 | 26.89 | 929.51 | 1461.60 |

The difference becomes more pronounced on medium- to large-scale datasets. For SWaT and Flood, our method finishes in tens of seconds, whereas baselines such as cLSTM, cRNN, DYNOTEARS, and TiMINo often require hundreds to thousands of seconds, and methods like MVGC, PCMCI+, and tsFCI regularly exceed the 3-hour timeout. On the largest datasets (Bavaria and East Germany),

Table 21: Runtime (in seconds) for All Datasets with Lag = 10. TLE: time limit exceeded (3 hours).

| Method | MoM 1 | MoM 2 | Storm | Web 1 | Web 2 | Antivirus 1 | Antivirus 2 | SWaT | Flood | Bavaria | East Germany |
|---|---|---|---|---|---|---|---|---|---|---|---|
| VAR | 0 | 0.00 | 0.00 | 0.03 | 0.03 | 0.03 | 0.05 | 0.43 | 0.43 | 37.35 | 67.78 |
| MVGC | 0.209 | 0.35 | 0.19 | 4.52 | 4.42 | 4.83 | 5.24 | 2064.97 | 1265.48 | TLE | TLE |
| cLSTM | 34.63 | 38.22 | 51.91 | 125.74 | 165.95 | 106.02 | 94.09 | TLE | 656.06 | TLE | TLE |
| cMLP | 12.724 | 14.41 | 14.86 | 47.48 | 47.87 | 29.79 | 21.95 | 359.23 | 204.77 | TLE | TLE |
| cRNN | 19.313 | 21.27 | 29.25 | 167.09 | 129.71 | 66.18 | 66.44 | 6281.79 | 797.46 | TLE | TLE |
| TCDF | 8.69 | 9.02 | 9.94 | 58.36 | 58.55 | 16.65 | 17.81 | 550.15 | 258.03 | TLE | TLE |
| PCMCI+ | 0.619 | 1.33 | 7.67 | 231.03 | 81.34 | 39.42 | 39.85 | TLE | TLE | TLE | TLE |
| tsFCI | 93.79 | 53.46 | 1.55 | 549.18 | 3582.58 | 3179.41 | TLE | TLE | TLE | TLE | TLE |
| PCGCE | 0.299 | 0.61 | 1.46 | 9.09 | 5.62 | 38.82 | 10.85 | 856.84 | 81.11 | TLE | TLE |
| DYNOTEARS | 11.554 | 12.46 | 0.30 | 50.85 | 365.70 | 54.52 | 306.68 | 3086.13 | 516.16 | TLE | TLE |
| VARLINGAM | 0.320 | 0.34 | 0.59 | 8.94 | 9.44 | 1.30 | 1.24 | 2945.26 | 55.11 | TLE | TLE |
| TiMINo | 1.468 | 1.48 | 2.34 | 14.98 | 13.57 | 7.00 | 7.09 | 636.87 | 78.71 | TLE | TLE |
| Our Method | 7.168 | 8.60 | 7.41 | 12.43 | 16.38 | 7.61 | 7.04 | 33.78 | 18.01 | 1498.22 | 2521.38 |

nearly all baseline methods fail to return results within the time limit, while our method is able to complete in under one hour to about forty minutes, depending on the lag setting.

An additional observation is that while deep learning-based methods (cLSTM, cRNN, cMLP) often scale poorly and time out on larger datasets, our approach remains robust across different lags. Moreover, causal discovery methods that explicitly enforce acyclicity (e.g., DYNOTEARS, VAR-LiNGAM, TiMINo) typically exhibit severe runtime growth, while our method avoids this by embedding acyclicity into a differentiable reparameterization, allowing for end-to-end optimization without expensive combinatorial operations.

## A.9 ADDITIONAL GPU RUNTIME RESULTS

Across all three representative subsets (Web 2, SWaT, and Flood) and with lag set to 3, our method consistently achieves the lowest GPU runtime among all competing approaches.

Table 22: GPU runtime (in seconds) with Lag=3

| Dataset | cLSTM | cMLP | cRNN | TCDF | Our Method |
|---|---|---|---|---|---|
| Web 2 | 56.845376 | 24.166523 | 48.599175 | 27.212114 | 8.754724 |
| SWaT | 124.474320 | 72.848101 | 113.851332 | 386.728746 | 16.438581 |
| Flood | 109.431931 | 74.310966 | 100.446660 | 100.163351 | 13.656314 |

Table 22 shows that neural Granger-causality–based baselines (TCDF, cLSTM, cMLP, and cRNN) are substantially slower, even when accelerated by GPU hardware. This is primarily due to their recurrent or deep feed-forward architectures, which require multiple sequential GPU kernels per update step. TCDF also incurs considerable GPU cost because of its convolutional architecture and attention-based occlusion evaluation.

In contrast, our method maintains a lightweight computational structure: the optimization relies on closed-form operations over low-dimensional matrices, without recurrent unrolling or deep convolutional layers. As a result, GPU parallelism is fully utilized with minimal overhead.

On all datasets, our method is 5×–10× faster than cMLP/cLSTM/cRNN and 20×+ faster than TCDF on SWaT. These results demonstrate that even under GPU acceleration, our method retains a clear efficiency advantage over neural-network-based causal discovery baselines.

## A.10 ANALYSIS FOR SEPARATE INSTANTANEOUS AND LAGGED EFFECTS

To better understand how the proposed method handles instantaneous and lagged dependencies, we conduct a detailed case study on the MoM1 dataset under a maximum lag of three. The MoM1 dataset records the behaviour of a publish-subscribe based message middleware and includes seven system-level variables such as publish rate, number of consumers, queued messages, CPU usage, RAM usage, disk read, and disk write. The expert-defined ground truth causal graph, which reflects the functional structure of the middleware system, is shown in Figure 2.

We evaluate the inferred causal structure at each lag from zero to three. The quantitative performance is summarised in Table 23. The causal graphs corresponding to the four lag settings are shown in Figure 3, where each subplot illustrates the directed dependencies inferred by the model when only the specified lag is considered.

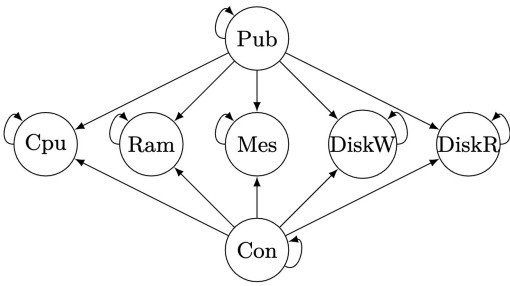

Figure 2: Ground truth causal graph for the MoM1 dataset.

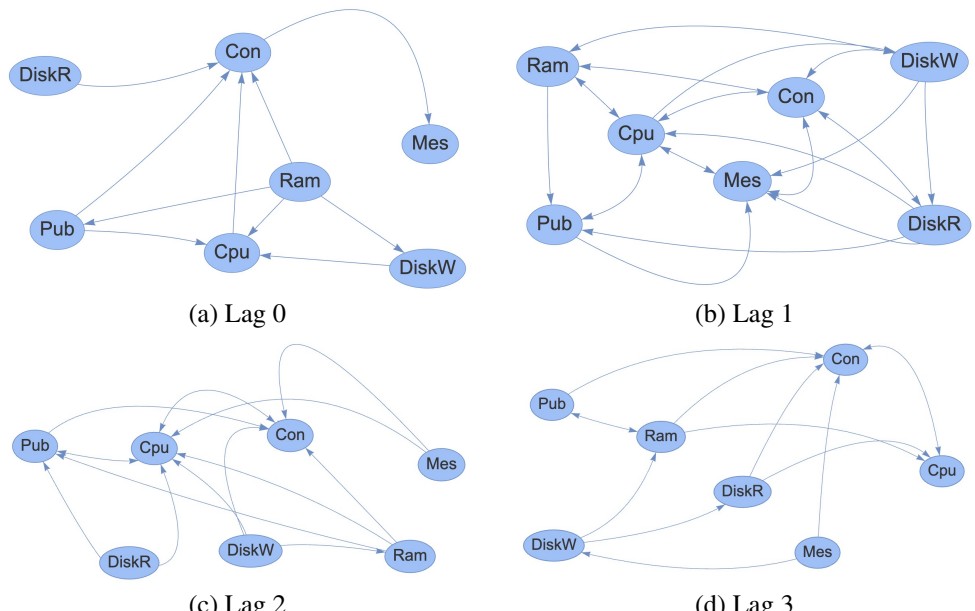

(a) Lag 0

(b) Lag 1

(c) Lag 2

(d) Lag 3

Figure 3: Inferred causal graphs for the MoM1 dataset under different lag settings.

Table 23: Performance across different lags on MOM1 dataset

| Lag | F1 | Precision | Recall |
| --- | --- | --- | --- |
| 0 | 0.200 | 0.200 | 0.200 |
| 1 | 0.412 | 0.292 | 0.700 |
| 2 | 0.160 | 0.133 | 0.200 |
| 3 | 0.174 | 0.154 | 0.200 |

The trend shown in Table 23 indicates that lag one achieves the best performance with an F1 score of 0.412, notably higher than the other three settings. Lag zero results in more instantaneous connections among system metrics but suffers from reduced distinguishability between true causal effects and correlations arising from shared system load, yielding lower precision. Lags two and three produce many long-range dependencies that are unlikely to represent meaningful causal effects given the short reaction time of the middleware system, resulting in low scores.

These observations align with the characteristics of the MoM1 system. The publish and consume rates influence the queued messages almost immediately due to the high sampling frequency and the

low-latency nature of the middleware. As a result, instantaneous effects and short lag-one effects dominate the true causal structure. This explains why lag one recovers the most coherent subgraph among the four settings and why longer lags introduce spurious edges unrelated to the system dynamics.

Overall, the results suggest that the method is able to capture short-term causal influences more reliably than long-term dependencies in this dataset. The instantaneous and lag-one effects are the most meaningful for MoM1, and these are the lags where the proposed method most accurately reflects the ground truth structure. This analysis demonstrates that separating instantaneous and lagged evaluations helps reveal how causal influence is distributed over time and highlights the importance of selecting an appropriate maximum lag for low-latency IT monitoring systems.

### A.11 SENSITIVITY ANALYSIS

In this section, we analyze the sensitivity of our method to different hyperparameter settings, including the pruning threshold, the sparsity coefficients of instantaneous and lagged effects, and the random seed. We conduct all experiments on two representative datasets, MoM1 and Web1, using a maximum lag of three. This section summarizes how the method behaves under different parameter choices and discusses the implications of these variations.

#### A.11.1 SENSITIVITY TO PRUNING THRESHOLD

Table 24 reports the performance of our method under different pruning thresholds on the MoM1 and Web1 datasets (Lag = 3). The results show a clear trend: small thresholds (e.g., 0.001–0.01) generally produce higher recall but may retain more spurious edges, whereas larger thresholds (e.g., 0.05–0.7) suppress weak connections aggressively and cause recall to drop. The F1 score follows a non-monotonic pattern, peaking at moderate thresholds depending on the dataset.

Table 24: Results under different pruning thresholds (Lag = 3)

| Dataset | Metric | 0.001 | 0.002 | 0.005 | 0.007 | 0.01 | 0.02 | 0.05 | 0.07 | 0.1 | 0.5 | 0.7 |
|---------|--------|-------|-------|-------|-------|------|------|------|------|-----|-----|-----|
| MoM1 | F1 | 0.3846 | 0.4255 | 0.4091 | 0.4186 | 0.4000 | 0.3429 | 0.3704 | 0.2400 | 0.1905 | 0.1538 | 0.1667 |
| | Precision | 0.2381 | 0.2703 | 0.2647 | 0.2727 | 0.2667 | 0.2400 | 0.2941 | 0.2000 | 0.1818 | 0.3333 | 0.5000 |
| | Recall | 1.0000 | 1.0000 | 0.9000 | 0.9000 | 0.8000 | 0.6000 | 0.5000 | 0.3000 | 0.2000 | 0.1000 | 0.1000 |
| Web1 | F1 | 0.2826 | 0.3023 | 0.2892 | 0.3117 | 0.3243 | 0.3030 | 0.2917 | 0.3077 | 0.1250 | 0.1111 | 0.1111 |
| | Precision | 0.1667 | 0.1806 | 0.1739 | 0.1905 | 0.2000 | 0.1923 | 0.2059 | 0.2400 | 0.1111 | 0.2500 | 0.2500 |
| | Recall | 0.9286 | 0.9286 | 0.8571 | 0.8571 | 0.8571 | 0.7143 | 0.5000 | 0.4286 | 0.1429 | 0.0714 | 0.0714 |

For MoM1, the best F1 score is achieved near thresholds 0.002–0.007, where recall remains high but precision also begins to improve. For Web1, a similar pattern is observed, with moderate pruning producing the strongest balance between precision and recall. These observations indicate that pruning threshold plays a central role in regulating sparsity: when set too low, the graph becomes noisy; when set too high, true causal edges are prematurely removed.

This finding aligns with our broader sensitivity analysis and is consistent with the behaviour observed on the AntiV1/AntiV2 datasets, discussed in Appendix A.5. In the AntiVirus case, the performance degradation seen in the main experiments is largely due to the fixed pruning threshold of 0.01 not being well matched to the irregular and bursty dynamics of those datasets. Once the threshold is tuned, performance improves substantially. Together, these results show that while our method is effective, its performance can be sensitive to the choice of pruning threshold, especially for datasets with heterogeneous or highly irregular temporal structures.

#### A.11.2 SENSITIVITY TO SPARSITY COEFFICIENTS $\lambda_0$ AND $\lambda_\tau$

We further study the sensitivity of our method to the sparsity coefficients applied to instantaneous effects ($\lambda_0$) and lagged effects ($\lambda_\tau$). These hyperparameters control the strength of the $\ell_1$ penalty and therefore directly influence how aggressively the estimated causal matrices are sparsified. We evaluate three configurations on MoM1 and Web1 with maximum lag equal to three: (i) varying $\lambda_0$ and $\lambda_\tau$ simultaneously, (ii) fixing $\lambda_0$ while varying $\lambda_\tau$, and (iii) fixing $\lambda_\tau$ while varying $\lambda_0$. The results are reported in Tables 25, 26, and 27.

Table 25: Results under different values of $\lambda_0$ and $\lambda_\tau$, varying simultaneously (lag = 3)

| Dataset | Metric | 0.0 | 0.0002 | 0.0005 | 0.001 | 0.002 | 0.005 | 0.01 | 0.05 | 0.1 | 0.5 |
|---|---|---|---|---|---|---|---|---|---|---|---|
| | F1 | 0.3922 | 0.3846 | 0.4000 | 0.4000 | 0.3077 | 0.2000 | 0.2222 | 0.0000 | 0.0000 | 0.0000 |
| MoM1 | Precision | 0.2439 | 0.2381 | 0.2571 | 0.2667 | 0.2069 | 0.2000 | 0.2500 | 0.0000 | 0.0000 | 0.0000 |
| | Recall | 1.0000 | 1.0000 | 0.9000 | 0.8000 | 0.6000 | 0.2000 | 0.2000 | 0.0000 | 0.0000 | 0.0000 |
| | F1 | 0.2857 | 0.2857 | 0.3291 | 0.3243 | 0.3200 | 0.2927 | 0.1935 | 0.2222 | 0.0000 | 0.0000 |
| Web1 | Precision | 0.1688 | 0.1688 | 0.2000 | 0.2000 | 0.1967 | 0.2222 | 0.1765 | 0.5000 | 0.0000 | 0.0000 |
| | Recall | 0.9286 | 0.9286 | 0.9286 | 0.8571 | 0.8571 | 0.4286 | 0.2143 | 0.1429 | 0.0000 | 0.0000 |

Table 26: Results with fixed $\lambda_0 = 0.001$ and varying $\lambda_\tau$ (lag = 3)

| Dataset | Metric | 0.0 | 0.0002 | 0.0005 | 0.001 | 0.002 | 0.005 | 0.01 | 0.05 | 0.1 | 0.5 |
|---|---|---|---|---|---|---|---|---|---|---|---|
| | F1 | 0.3846 | 0.3846 | 0.4167 | 0.4000 | 0.3889 | 0.1765 | 0.1818 | 0.2143 | 0.4000 | 0.1935 |
| MoM1 | Precision | 0.2381 | 0.2381 | 0.2632 | 0.2667 | 0.2692 | 0.1250 | 0.1304 | 0.1667 | 0.3000 | 0.1429 |
| | Recall | 1.0000 | 1.0000 | 1.0000 | 0.8000 | 0.7000 | 0.3000 | 0.3000 | 0.3000 | 0.6000 | 0.3000 |
| | F1 | 0.2955 | 0.3059 | 0.3038 | 0.3243 | 0.2941 | 0.2985 | 0.1818 | 0.2500 | 0.2326 | 0.2727 |
| Web1 | Precision | 0.1757 | 0.1831 | 0.1846 | 0.2000 | 0.1852 | 0.1887 | 0.1220 | 0.1923 | 0.1724 | 0.2000 |
| | Recall | 0.9286 | 0.9286 | 0.8571 | 0.8571 | 0.7143 | 0.7143 | 0.3571 | 0.3571 | 0.3571 | 0.4286 |

Table 27: Results with fixed $\lambda_\tau = 0.001$ and varying $\lambda_0$ (lag = 3)

| Dataset | Metric | 0.0 | 0.0002 | 0.0005 | 0.001 | 0.002 | 0.005 | 0.01 | 0.05 | 0.1 | 0.5 |
|---|---|---|---|---|---|---|---|---|---|---|---|
| | F1 | 0.4167 | 0.4082 | 0.3415 | 0.4000 | 0.3636 | 0.1935 | 0.2143 | 0.2069 | 0.1935 | 0.2500 |
| MoM1 | Precision | 0.2632 | 0.2564 | 0.2258 | 0.2667 | 0.2609 | 0.1429 | 0.1667 | 0.1579 | 0.1429 | 0.1818 |
| | Recall | 1.0000 | 1.0000 | 0.7000 | 0.8000 | 0.6000 | 0.3000 | 0.3000 | 0.3000 | 0.3000 | 0.4000 |
| | F1 | 0.2824 | 0.2857 | 0.3200 | 0.3243 | 0.2927 | 0.3200 | 0.2581 | 0.2623 | 0.2667 | 0.2540 |
| Web1 | Precision | 0.1690 | 0.1714 | 0.1967 | 0.2000 | 0.1765 | 0.1967 | 0.1667 | 0.1702 | 0.1739 | 0.1633 |
| | Recall | 0.8571 | 0.8571 | 0.8571 | 0.8571 | 0.8571 | 0.8571 | 0.5714 | 0.5714 | 0.5714 | 0.5714 |

Across both datasets, we observe a consistent pattern indicating that the sparsity penalty interacts strongly with the optimization dynamics of the permutation-based SVAR model. When both $\lambda_0$ and $\lambda_\tau$ increase simultaneously, performance is relatively stable for small coefficients (up to around 0.001), after which both precision and recall drop sharply. For MoM1, the F1 score collapses to zero once the sparsity coefficients reach 0.05 or higher, reflecting that overly strong penalties prune nearly all edges. A similar trend is observed on Web1. This behavior is expected because the Sinkhorn model represents causal strengths in a linear SVAR structure, where weak but meaningful coefficients can be easily suppressed when the sparsity penalty becomes too dominant.

When isolating the effect of lagged sparsity by fixing $\lambda_0 = 0.001$, we observe a more gradual decline in performance as $\lambda_\tau$ increases. For MoM1, moderate values (0.0005–0.002) yield comparable F1 scores, suggesting that lagged coefficients have a wider tolerance range before being pruned excessively. For Web1, the performance remains relatively stable at lower $\lambda_\tau$ values and only begins to degrade when $\lambda_\tau$ exceeds 0.01. This indicates that the lagged component of the model is somewhat more robust to sparsity penalization.

A different behavior emerges when fixing $\lambda_\tau = 0.001$ and varying $\lambda_0$. The instantaneous sparsity coefficient has a stronger influence on model performance. For MoM1, the best F1 occurs at $\lambda_0 = 0$ or very small values, and performance degrades significantly once $\lambda_0$ exceeds 0.005. This is consistent with the fact that instantaneous effects in MoM1 are relatively weak compared to lagged dependencies, so applying high sparsity directly on $B_0$ removes essential causal links. A similar trend holds for Web1, where instantaneous sparsity must remain small to avoid suppressing meaningful instantaneous interactions.

Overall, these results highlight a common conclusion: the Sinkhorn method is sensitive to sparsity hyperparameters and benefits from small $\lambda_0$ and $\lambda_\tau$ values, particularly for datasets where causal effects are subtle or of small magnitude. Excessive sparsity quickly leads to underfitting, eliminating key edges and collapsing recall. This analysis is consistent with our observations on pruning thresh-

old sensitivity (Sec. A.11.1) and the AntiV datasets (Appendix A.5), all of which indicate that while sparsity controls are useful, they must be chosen carefully to avoid erasing true causal relationships.

### A.11.3 SENSITIVITY TO RANDOM SEED

Table 28 reports the performance of our method under different random seeds on the MoM1 and Web1 datasets (lag = 3). The results show moderate but noticeable fluctuations across seeds. On MoM1, the F1 score ranges from 0.2857 to 0.4000, while on Web1 it varies between 0.2500 and 0.3243. Precision and recall exhibit similar levels of variability. Although the performance remains within a relatively stable band, these differences demonstrate that the method is not entirely deterministic even when all other hyperparameters are fixed.

Table 28: Results under different random seeds (lag = 3)

| Dataset | Metric | 1 | 7 | 42 | 77 | 123 | 999 |
|---------|--------|------|------|------|------|------|------|
| **MoM1** | F1 | 0.3125 | 0.3684 | 0.4000 | 0.3784 | 0.2857 | 0.3810 |
| | Precision | 0.2273 | 0.2500 | 0.2667 | 0.2593 | 0.2000 | 0.2500 |
| | Recall | 0.5000 | 0.7000 | 0.8000 | 0.7000 | 0.5000 | 0.8000 |
| **Web1** | F1 | 0.2667 | 0.3117 | 0.3243 | 0.2500 | 0.3200 | 0.3117 |
| | Precision | 0.1639 | 0.1905 | 0.2000 | 0.1552 | 0.1967 | 0.1905 |
| | Recall | 0.7143 | 0.8571 | 0.8571 | 0.6429 | 0.8571 | 0.8571 |

This behaviour arises naturally from several components of the model. First, the optimization relies on stochastic initialization of both the instantaneous coefficient matrix and the permutation logits, which determine the causal ordering via the Gumbel–Sinkhorn relaxation. These initial conditions can steer the optimization trajectory toward different local minima, especially in non-convex landscapes. Second, the use of Gumbel noise in the relaxation process injects additional randomness during training, impacting the learned permutation matrix and thus the structure of the estimated causal graph. Third, the stochastic nature of gradient-based optimization, including Adam's internal state and floating-point ordering on different hardware, may lead to slight variations in convergence paths even with fixed seeds.

Despite these sources of randomness, the fluctuations observed in Table 28 remain within a bounded range. Importantly, the qualitative structure of the learned causal graphs stays consistent across seeds, with core causal pathways repeatedly recovered. This indicates that while stochasticity influences the fine-grained details of edge weights and pruning outcomes, the overall causal patterns remain robust. These observations also highlight the importance of reporting results across multiple seeds or adopting stability-selection strategies in future extensions of the method.

### A.12 SYNTHETIC DATA ANALYSIS

While the experiments in the main paper focus on real-world datasets, synthetic data allow for controlled studies of how our method behaves under different generative conditions. In this section, we evaluate (i) distributional robustness under different noise assumptions, (ii) sensitivity to sparsity levels, and (iii) the effect of varying the true maximum lag. All synthetic datasets contain 30 variables and are generated from a structural VAR model with instantaneous effects (matrix $B_0$) and lagged effects $\{B_\tau\}_{\tau=1}^{L}$. The ground-truth adjacency matrix is derived from the union of all non-zero entries across $B_0$ and the lagged matrices.

### A.12.1 SYNTHETIC DATA GENERATED UNDER DIFFERENT DISTRIBUTIONAL ASSUMPTIONS

To examine robustness to deviations from Gaussianity, which is an assumption implicitly used by many linear causal discovery methods, we generated six synthetic datasets corresponding to different noise distributions while keeping the same underlying causal structure. The noise types include Gaussian noise, Laplace noise with heavier tails, Student-T noise with moderate degrees of freedom, Uniform noise with bounded support that violates typical moment assumptions, a mixture of Gaussians representing a bimodal and non-symmetric distribution, and a skewed distribution introducing non-zero asymmetry. For each dataset, we fixed the true maximum lag at $L = 3$, used the same instantaneous and lagged sparsity levels, and evaluated our method with the default configuration.

Table 29: Results on synthetic data under different distributional assumptions

| Noise | Gaussian | Laplace | StudentT | Uniform | MixGauss | Skewed |
|---|---|---|---|---|---|---|
| F1 | 0.8593 | 0.8569 | 0.8565 | 0.5993 | 0.8492 | 0.8670 |
| Precision | 0.7920 | 0.7758 | 0.7860 | 0.6743 | 0.7850 | 0.7928 |
| Recall | 0.9391 | 0.9570 | 0.9409 | 0.5392 | 0.9248 | 0.9565 |

Table 29 summarizes the performance under these six noise conditions. The results show that the method performs consistently well across Gaussian, Laplace, Student-T, Mixture of Gaussians, and Skewed noise. In all these cases, the F1 scores remain in the range of approximately $0.85$ to $0.87$. This indicates that the method is robust to heavy-tailed distributions, asymmetric distributions, and multi-modal noise structures. The recall values are also high for these settings, suggesting that the model is able to reliably recover true causal edges even when the noise deviates substantially from Gaussianity.

In contrast, Uniform noise leads to a pronounced drop in performance, with the F1 score decreasing to around $0.60$. This behavior is expected, since bounded Uniform noise lacks informative tail structure and produces weaker statistical gradients during optimization, which makes it more challenging to detect subtle causal dependencies. The recall falls to $0.54$, indicating that the method misses a noticeable fraction of true edges under this noise condition.

Interestingly, the skewed noise distribution yields the highest F1 score among all settings. In this case, the skewness introduces additional linear dependencies that appear to make the causal structure easier to recover, while still preserving a favorable signal-to-noise ratio. Taken together, these results demonstrate that the method is broadly robust to a wide class of non-Gaussian disturbances, and struggles primarily in situations where the noise is bounded and lacks tail information, as in the Uniform case.

### A.12.2 SYNTHETIC DATA GENERATED UNDER DIFFERENT SPARSITY LEVELS

To examine how the proposed method behaves when the underlying causal graph becomes more or less connected, we generated six datasets with increasing sparsity levels, ranging approximately from $0.5\%$ to $8\%$ of all possible edges. The maximum lag was fixed to three and the instantaneous and lagged edge probabilities were adjusted to produce graphs of different densities. We evaluated the method under three regularization settings: the default values $\lambda_0 = \lambda_\tau = 0.001$, a smaller regularization strength ($\lambda_0 = \lambda_\tau = 0.0001$), and a larger one ($\lambda_0 = \lambda_\tau = 0.002$). Tables 30, 31, and 32 summarize the results.

Table 30: Results on synthetic data under different sparsity levels ($\lambda_0$ and $\lambda_\tau$ are using default 0.001)

| Sparsity | 0.5% | 1% | 2% | 3% | 5% | 8% |
|---|---|---|---|---|---|---|
| F1 | 0.5746 | 0.7584 | 0.8072 | 0.8493 | 0.4244 | 0.5514 |
| Precision | 0.4148 | 0.6181 | 0.7002 | 0.7855 | 0.9163 | 0.9940 |
| Recall | 0.9348 | 0.9811 | 0.9530 | 0.9244 | 0.2762 | 0.3815 |

Table 31: Results on synthetic data under different sparsity levels (both $\lambda_0$ and $\lambda_\tau$ fixed to 0.0001)

| Sparsity | 0.5% | 1% | 2% | 3% | 5% | 8% |
|---|---|---|---|---|---|---|
| F1 | 0.3009 | 0.4897 | 0.7056 | 0.8411 | 0.6553 | 0.7872 |
| Precision | 0.1775 | 0.3251 | 0.5471 | 0.7258 | 0.9222 | 0.9930 |
| Recall | 0.9855 | 0.9924 | 0.9936 | 1.0000 | 0.5082 | 0.6520 |

The default configuration exhibits a clear trend. Performance improves steadily from extremely sparse to moderately sparse graphs, with the F1 score increasing from approximately $0.57$ at $0.5\%$ sparsity to a peak of approximately $0.85$ at $3\%$. Beyond this point, as the graphs become denser, the F1 score drops noticeably, reaching $0.42$ at $5\%$ and $0.55$ at $8\%$. The precision–recall profile illustrates why this happens. For very sparse graphs, recall is high but precision is low: the method

Table 32: Results on synthetic data under different sparsity levels (both $\lambda_0$ and $\lambda_\tau$ fixed to 0.002)

| **Sparsity** | 0.5% | 1% | 2% | 3% | 5% | 8% |
|---|---|---|---|---|---|---|
| F1 | 0.8151 | 0.8700 | 0.8268 | 0.8076 | 0.3657 | 0.4406 |
| Precision | 0.7727 | 0.8310 | 0.7664 | 0.7956 | 0.9188 | 0.9919 |
| Recall | 0.8623 | 0.9129 | 0.8974 | 0.8199 | 0.2282 | 0.2832 |

detects most true edges but also produces many false positives. In contrast, for extremely dense graphs such as the 8% case, precision becomes very high whereas recall drops sharply. This indicates that when the ground truth contains many edges, the regularizer encourages overly sparse solutions, suppressing true dependencies and reducing recall. Overall, the method performs best in the intermediate region where the true structure is neither extremely sparse nor highly connected.

When decreasing the regularization strength to $\lambda_0 = \lambda_\tau = 0.0001$, recall becomes extremely high across all sparsity levels, often reaching values above 0.99, but precision drops substantially in the sparser regimes. This results in markedly lower F1 scores for the 0.5% and 1% cases. This behaviour can be interpreted as under-regularization: with weaker shrinkage, the model tends to retain many spurious coefficients, especially when the true graph is small. However, this setting performs comparatively well for moderately sparse graphs: for instance, the F1 score at 3% sparsity reaches approximately 0.84, close to the default-value performance. This suggests that the benefit of a weaker sparsity penalty depends strongly on the true underlying density.

In contrast, increasing the regularization strength to $\lambda_0 = \lambda_\tau = 0.002$ yields an almost opposite effect. Precision remains consistently high across all sparsity levels, but recall is suppressed, especially in the denser cases. For example, at 5% sparsity, recall falls to approximately 0.23, and at 8% sparsity it drops to approximately 0.28. This corresponds to an over-regularized regime in which many true edges are removed. Interestingly, for very sparse graphs the stronger penalty improves performance noticeably. At 0.5% and 1% sparsity the F1 scores reach approximately 0.82 and 0.87, respectively, substantially higher than the default and smaller-lambda results. This indicates that stronger regularization is helpful precisely when the true causal structure is extremely sparse.

Taken together, these experiments illustrate that the sparsity regularizer is not universally beneficial across all settings but instead interacts with the true density of the underlying system. This echoes concerns raised in peer review that sparsity is domain dependent and should not be treated as a universal property of causal structures. Real-world systems can be moderately or highly connected, and in such cases, enforcing strong sparsity can remove true dependencies and degrade recovery accuracy by suppressing recall. Conversely, when the true causal graph is very sparse, stronger regularization substantially improves precision and overall accuracy. These observations highlight the importance of adaptively tuning the sparsity penalty based on prior knowledge of system connectivity or using data-driven strategies to determine an appropriate regularization level.

### A.12.3 SYNTHETIC DATA ANALYSIS UNDER LAG MISMATCH

To understand how the method behaves when the maximum temporal lag used by the model does not match the true underlying lag of the data-generating process, we evaluate the method on synthetic datasets generated with lag values of 3, 5, and 10, and run the model using maximum lags of 3, 5, and 10. The results are presented in Table 33.

The first observation is that when the true lag is small, for example equal to 3, the model performs consistently well regardless of whether the chosen model lag is equal to, slightly larger than, or substantially larger than the true lag. All three settings produce similar F1 and recall values. This indicates that increasing the model lag beyond the true value does not introduce many spurious edges. The model is able to automatically ignore lagged coefficients that are not supported by the data and maintain a stable causal structure. This robustness is desirable, as it shows that overestimating the lag does not severely degrade the quality of the learned graph.

In contrast, when the true underlying lag is larger than the model lag, performance decreases significantly. For example, when the true lag is 5 or 10 but the model is restricted to lag 3, both F1 and recall drop sharply. This behaviour is expected: a model capped at lag 3 is fundamentally unable to capture causal effects that occur at lag 4 or beyond. Any true relationships that manifest

Table 33: Performance under different combinations of true lag (generation) and model maximum lag.

| Dataset | ModelLag | F1 | Precision | Recall |
|---------|----------|-----|-----------|--------|
| Lag3 (generated with Lag 3) | 3 | 0.8576 | 0.8125 | 0.9079 |
| | 5 | 0.8512 | 0.8048 | 0.9032 |
| | 10 | 0.8571 | 0.8105 | 0.9095 |
| Lag5 (generated with Lag 5) | 3 | 0.3448 | 0.8387 | 0.2170 |
| | 5 | 0.4129 | 0.8250 | 0.2754 |
| | 10 | 0.4420 | 0.8051 | 0.3046 |
| Lag10 (generated with Lag 10) | 3 | 0.4774 | 0.9853 | 0.3150 |
| | 5 | 0.5185 | 0.9805 | 0.3525 |
| | 10 | 0.5585 | 0.9766 | 0.3911 |

only at higher lags cannot be recovered, which directly lowers recall and consequently the F1 score. This demonstrates the importance of choosing a model lag that is at least as large as the longest meaningful temporal dependency in the system.

A more subtle pattern emerges when focusing on cases where the true lag is large. Even when the model lag is increased to match or exceed the true value, the improvement in performance is modest rather than dramatic. For example, when the true lag is 5, increasing the model lag from 3 to 5 yields only limited gains, and the increase from 5 to 10 produces only a small further improvement. A similar pattern appears for data generated with lag 10. This reflects the fact that learning long-range dependencies is intrinsically harder. Higher lag orders substantially expand the parameter space and increase statistical difficulty, making it more challenging to reliably infer distant lagged edges even when the model is given the correct lag. This difficulty is especially pronounced in high-dimensional settings, where the number of possible temporal edges grows quadratically with the number of variables and linearly with the lag order.

Taken together, these results highlight three key points. First, the method is robust to overestimation of lag, as using a model lag that exceeds the true value does not significantly degrade performance. Second, underestimation of the lag causes consistent and substantial loss of recall because true lagged edges cannot be represented by the model. Third, even when the lag is correctly specified, causal discovery becomes increasingly difficult as the true lag grows, because the underlying temporal structure becomes more complex and requires more samples and stronger regularization to recover reliably. These findings suggest that in practice it is safer to err on the side of a slightly larger model lag, while also recognizing that very long-range causal dependencies remain challenging to detect even under ideal conditions.

### A.12.4 SUMMARY AND DISCUSSION

Although synthetic data provide controlled settings for probing sparsity, noise characteristics, and lag mismatch, their absolute performance values are noticeably higher than those obtained on real-world datasets in the main paper. This gap highlights a key point: synthetic data are inherently easier, as their structural assumptions, noise processes, and temporal dependencies are clean, well behaved, and closely aligned with the conditions typically assumed by causal discovery algorithms. Consequently, methods often appear more accurate on synthetic benchmarks than when faced with the complexity, heterogeneity, and irregular dynamics of real-world systems.

Real-world datasets therefore offer a more stringent and meaningful test of practical utility. They capture distributional shifts, latent confounding, irregular sampling, and non-stationarity—factors that algorithms must confront in realistic applications. Synthetic datasets, by contrast, excel in a complementary role. They are well suited for sensitivity analyses, controlled ablations, and diagnostic studies aimed at understanding how modeling choices—such as lag selection, sparsity regularization, or robustness to non-Gaussian noise—affect algorithmic behavior. Their precisely manipulable structure allows isolation and evaluation of individual components in ways that real data cannot easily support.

