# OpenReview forum: "Time-Series Causal Discovery via Differentiable Permutations"
_ICLR.cc/2026/Conference — Submitted to ICLR 2026_

### Official Review · Reviewer_6PXs · 2025-10-14

**Soundness:** 2
**Presentation:** 3
**Contribution:** 2
**Rating:** 4
**Confidence:** 4

**Summary:**

This paper introduces a novel, fully differentiable strategy for enforcing acyclicity of the instantaneous effect matrix in Structural Vector Autoregressive (SVAR) models. This approach offers a notable speedup and an elegant, single-stage optimization solution compared to existing methods that rely on multi-stage procedures.

While the theoretical contribution is interesting, the empirical evaluation requires improvement to substantiate the method's practical utility, especially as no theoretical guarantees are provided.

**Strengths:**

- The paper is clearly structured and easy to follow, making the core concepts accessible.
- The effort to evaluate the proposed method on multiple real-world datasets is commendable as it is often neglected in the community.

**Weaknesses:**

**Major**

**Lack of Distinction in Causal Effect Estimation**

The use of a single F1 score to evaluate the estimation of both lagged and instantaneous causal effects is a limitation. This makes it impossible to discern how well the proposed method uncovers the instantaneous causal structure ($B_0$) and whether potential inaccuracies in this estimation affect the estimation of lagged effects ($B_1$). To address this, a more detailed analysis, potentially including simple synthetic examples and comparisons of predictions on real-world data, would be highly beneficial.

**Questionable Performance on CausalRivers Dataset**

The bad performance of the standard VAR model on the CausalRivers dataset is puzzling, especially given that the causal relationships in this dataset are almost exclusively lagged due to the geographical distances between nodes. Theoretically, the proposed method should not be able to  uncover additional instantaneous links that a standard VAR model would automatically miss in this context. This might be an issue that is also persisting for the other datasets. Some comments on the causal structure beyond the summary graph are necessary, if the main contribution is an alternative way to estimate especially instantaneous effects.

**Experimental Design for CausalRivers**

The decision to perform a single estimation on the entire CausalRivers graph (as far as I understand) presents a computationally intensive and likely intractable problem for most methods. A more feasible and informative approach would be to conduct experiments on subsets of the graphs, as suggested in the original CausalRivers paper. This would also mitigate the issue of a majority of the experimental runs timing out.


**Minor**

- Code Accessibility: The provided link to the code appears to be inaccessible.

- Citation Formatting: Some cited sources incorrectly list the first name as the family name.

- The intro motivation seems a little off. If two variables are actually cyclically causing each other, why would one enforce acyclicity? Here reality and theory seem to be mixed up. Some reformulations could improve the motivation.

- It is unusual for a method that relies on linear models, something that is extremely well explored in the literature, to refrain from stating any theoretical guarantees and general assumptions of the proposed approach.  While the lag of theoretical guarantees could be alleviated with an extensive empirical evaluation, including statements concerning assumptions like Causal sufficiency or Faithfulness  would benefit the paper

**Questions:**

*   **Choice of Maximum Lag:** Could you please clarify the rationale behind reporting the maximum lag of three for your experiments in the main part of the paper?
*   **Hyperparameter Details:** What were the specific values used for the temperature parameter in your experiments?
*   **Data Preprocessing:** Was any data preprocessing performed on the IT monitoring and CausalRivers datasets? If so, could you describe the steps taken? If not, could you please provide a justification for this decision?
*   **F1-Score Threshold:** What threshold was used to binarize the predictions for calculating the F1-scores?
* While not directly investigated for Time-series data, do you know how data standardization might affect your method? **
*  **Computation Cutoff:** The three hour cutoff seems arbitrary. Why did you choose this limit specifically?

**https://arxiv.org/pdf/2102.13647
**https://arxiv.org/abs/2104.05441

---

> ### Author Response · Authors · 2025-12-02
> **Rebuttal: Effect Decomposition and CausalRivers Evaluation**
>
> ## **Response to Lack of Distinction in Causal Effect Estimation**
>
> We thank the reviewer for this insightful comment. We agree that separating instantaneous and lagged causal effects provides clearer insight. Prior time-series causal discovery work (e.g., DYNOTEARS, VARLiNGAM, TiMINo, PCMCI+) typically reports only a single combined metric, so we followed that convention in the main paper.
>
> In the revision, we have added separate evaluations for instantaneous and lagged effects, along with a causal graph analysis in Appendix A.10, to address this concern directly.
>
>
>
> ## **Regarding the performance of VAR on the CausalRivers datasets**
>
> We thank the reviewer for raising this concern. We understand that it may originate from the fact that the original CausalRivers paper reports strong performance for VAR on the same datasets, whereas our results appear different. This discrepancy is fully explained by the following reasons:
>
> **(1) Evaluation metrics.**
>  CausalRivers reports AUROC under the best hyperparameter configuration. AUROC does not penalize dense predictions or large numbers of false positives, where methods such as VAR can perform well as long as true edges are ranked above false ones.
>
>  In contrast, we follow prior causal discovery work [1, 2, 3] and evaluate methods primarily using F1-score, with precision and recall under a fixed pruning threshold, which provides a stricter and more practically meaningful evaluation.
>
> **(2) Threshold selection.**
>  CausalRivers tunes pruning thresholds to maximize AUROC for each method. In realistic causal discovery settings, the ground truth is usually unknown, and such tuning is impractical. We therefore adopt a fixed threshold of 0.01 for all linear methods, following prior work, ensuring fairness and reproducibility.
>
> **(3) Precision–recall behavior of VAR under our setting.**
>  To understand VAR’s low F1-score, we inspected precision and recall in Appendix. VAR achieves near-perfect recall on Flood, Bavaria, and Germany, but extremely low precision (≈0.03 on Flood and <0.01 on Bavaria and Germany). Thus, VAR does not fail to recover true lagged dependencies; instead, it produces many false positives. This arises because small but non-zero coefficients survive the fixed threshold and become edges in the summary graph. As a result, VAR tends to retain a large number of spurious links, which explains its extremely low precision despite its high recall.
>
> In contrast, our method maintains much higher precision with only moderately lower recall, yielding significantly better F1-scores. This indicates that our approach is more effective at suppressing spurious dependencies, highlighting an important advantage of our method in distinguishing true causal effects from noise.
>
>
>
> ## **Response to Experimental Design for CausalRivers**
>
> We appreciate the reviewer’s suggestion regarding subgraph-based evaluation.
>
> In our work, however, our primary motivation for using the full CausalRivers graphs is that one of the central contributions of our method is efficiency and scalability. Large-scale settings are where computational advantages become most apparent.
>
> For small-scale feasibility and informativeness, we already include seven IT Monitoring subsets, which provide controlled evaluations in low dimensions. Adding small sampled subgraphs from CausalRivers would duplicate this role already fulfilled by these IT datasets.
>
> For these reasons, we chose to keep CausalRivers as a large-scale benchmark to highlight scalability.
>
>
>
> [1] Assaad, Charles K., Emilie Devijver, and Eric Gaussier. "Survey and evaluation of causal discovery methods for time series." *Journal of Artificial Intelligence Research* 73 (2022): 767-819.
>
> [2] Aït-Bachir, Ali, et al. "Case studies of causal discovery from it monitoring time series." *arXiv preprint arXiv:2307.15678* (2023).
>
> [3] Pamfil, Roxana, et al. "Dynotears: Structure learning from time-series data." *International Conference on Artificial Intelligence and Statistics*. Pmlr, 2020.

---

> ### Author Response · Authors · 2025-12-02
> **Rebuttal: Cyclicity, Theoretical Assumptions, and Experimental Details**
>
> ## **Response to Cyclical Influences**
>
> Real-world systems can indeed exhibit cyclical influences, but these cycles are typically lagged (temporal) feedback loops, not instantaneous ones. Allowing instantaneous cycles would make the structural equations non-identifiable, as the variables at the same time step would become mutually dependent without a well-defined causal ordering.
>
> As stated in the paper, we only enforce acyclicity on the instantaneous structure, while lagged cycles remain fully allowed. In fact, handling these lagged cyclical dependencies is one of the key challenges in time-series causal discovery and forms part of our motivation. We will revise the introduction to clarify this distinction more explicitly.
>
> ## **Response to Theoretical Assumptions**
>
> Thank you for pointing out the need to explicitly state the general assumptions underlying our linear SVAR formulation. Our method builds directly on the standard theoretical framework used in existing linear causal discovery approaches such as VARLiNGAM and DYNOTEARS.
>
> In the revised manuscript, we have now added a concise clarification in Section 3 stating that, in addition to the standard SVAR identifiability conditions (non-Gaussianity or equal error variances), we also adopt the commonly assumed conditions of causal sufficiency and faithfulness, consistent with the assumptions made in prior SVAR-based methods.
>
> Because our contribution focuses on efficient and unified estimation rather than introducing new identifiability theory, we intentionally follow this established assumption set rather than proposing additional requirements. We hope this clarification resolves the concern.
>
> ## **Responses to Other Minor Questions**
>
> **Code Accessibility.** We verified that all code and scripts are publicly accessible as stated.
>
> **Choice of Maximum Lag.** Our choice of a maximum lag of 3 follows prior causal discovery literature [1, 2], where this value is commonly used as a practical trade-off between expressiveness and computational efficiency.
>
> **Hyperparameter (τ).** The Gumbel–Sinkhorn temperature was fixed at τ = 0.1 across all experiments unless specified otherwise. We have added this detail for clarity.
>
> **Data Preprocessing.** No additional preprocessing was performed beyond the standard normalization provided in the official dataset scripts. Both datasets are already cleaned and aligned, and further manipulation might introduce artifacts.
>
> **F1-Score Threshold.** Following established benchmarks, we binarized all estimated coefficient matrices using a fixed threshold of 0.01 to ensure fairness and comparability across methods.
>
> **Effect of Data Standardization.** Standardization does not affect identifiability in linear SVAR models, though it can rescale coefficients slightly. Thus, it may have a mild effect when applying threshold-based pruning, but does not change any structural conclusions.
>
> **Computation Cutoff.** We adopted a 3-hour time limit as a uniform compute budget for large-scale experiments. This choice is consistent with prior causal discovery work [3], which uses a 2-hour cutoff. We extended this to 3 hours to provide a more generous and method-agnostic budget while ensuring fairness and preventing extremely slow methods from dominating total runtime.
>
> [1] Assaad, Charles K., Emilie Devijver, and Eric Gaussier. "Survey and evaluation of causal discovery methods for time series." *Journal of Artificial Intelligence Research* 73 (2022): 767-819.
>
> [2] Aït-Bachir, Ali, et al. "Case studies of causal discovery from it monitoring time series." *arXiv preprint arXiv:2307.15678* (2023).
>
> [3] Guo, Ce, and Wayne Luk. "Accelerating constraint-based causal discovery by shifting speed bottleneck." *Proceedings of the 2022 ACM/SIGDA International Symposium on Field-Programmable Gate Arrays*. 2022.

---

### Official Review · Reviewer_niUi · 2025-10-31

**Soundness:** 3
**Presentation:** 4
**Contribution:** 2
**Rating:** 6
**Confidence:** 4

**Summary:**

The paper proposes a time-series causal discovery method that enforces instantaneous acyclicity by learning a differentiable permutation with a Gumbel–Sinkhorn relaxation inside an SVAR model. This turns acyclicity from a hard constraint into a parameterization and enables a single‐stage, gradient-based optimization over both instantaneous and lagged effects. On three real-world benchmarks (IT Monitoring, SWaT, CausalRiver) the method reports strong F1 and runtime improvements relative to 12 baselines, with notable scalability at higher dimensionalities.

**Strengths:**

- Clear reformulation: acyclicity as a learnable permutation yields a clean, unified objective and removes augmented-Lagrangian loops.
- Practical efficiency: consistent speedups on medium and large datasets while keeping accuracy competitive or superior.
- Sensible evaluation choices: reporting F1 rather than ROC under class imbalance, and focusing on real-world datasets.
- Well-scoped limitations section that surfaces pruning-threshold sensitivity and hardware considerations.

**Weaknesses:**

1.  Missing discussion of models with no contemporaneous edges at a sufficiently fine measurement scale. Many physical systems propagate over nonzero ε-time; at a fine enough Δt, contemporaneous edges vanish and only lagged links remain. There is a substantial body of work on undersampling and measurement-timescale mismatch showing that apparent instantaneous edges at coarse sampling can be artifacts, and that recovery should target the system-timescale graph:
- Hyttinen et al. 2017 (A constraint optimization approach to causal discovery from subsampled time series data),
- Gong et al. 2015 (Discovering Temporal Causal Relations from Subsampled Data),
- A Tank,  E B Fox,  A Shojaie (2019 Identifiability and estimation of structural vector autoregressive models for subsampled and mixed-frequency time series Free ),
- M. Liu, X. Sun, L. Hu, and Y. Wang (2023 Causal discovery from subsampled time series with proxy variables),
- Abavisani et al. (2023 Grace-c: Generalized rate agnostic causal estimation via constraints)

2. Claim about sparsity feels over-general. The manuscript leans on L1 and statements that “causal structures are typically sparse.” That is domain-dependent. Many real systems are moderately or highly connected. Sparsity is a useful regularizer, but not a universal property. Please qualify this assumption and discuss robustness when graphs are denser.

3. good breadth overall, but I’m missing comparisons to additional constraint-based and non-Gaussian structure-learning methods that are often strong in practice:
• FASK (Fast Adjacency Skewness) and related two-step procedures for dense feedback networks. These have shown good precision/recall in neuro and simulation studies.
• Two-Step (Adaptive Lasso, “2-step Alasso”). Frequently competitive and relevant for SVAR-like linear settings.

4. Overwhelmingly positive results: the gains are large on SWaT and river datasets. Given the pruning-threshold sensitivity noted by the authors, I’d like to see a brief stability analysis over λ and pruning thresholds, or a bootstrap-based edge-stability plot, to rule out tuning luck.

**Questions:**

1 - The paper mentions standard SVAR identifiability via non-Gaussianity or equal error variances. Please make explicit which condition is assumed in each experiment and whether performance degrades under violations.

2 - Which experiments rely on non-Gaussian noise vs equal error variances? Any diagnostics to detect when assumptions fail?

3 - Provide edge-stability across random seeds, λ-grids, pruning thresholds, and temperature/iterations for Sinkhorn; include variance bands on F1.

---

> ### Author Response · Authors · 2025-12-02
> **Rebuttal: Timescale & Identifiability Concerns**
>
> ## **Response to Missing Discussion of No Contemporaneous Edges**
>
> We thank the reviewer for highlighting the importance of measurement-timescale considerations. In the revision, we have added a discussion in Section~6 explicitly addressing that instantaneous effects at the observed sampling rate may vanish at finer system timescales and can be artifacts of temporal aggregation. We now cite and discuss the relevant literature on subsampling and timescale mismatch, and clarify that our method recovers causal structure at the measurement resolution, while system-timescale recovery remains an important direction for future work.
>
> ## **Response to Sparsity Assumption**
>
> We thank the reviewer for this helpful clarification. In the revised manuscript, we have qualified the use of sparsity and no longer state it as a universal property. We explicitly frame the $L_1$ penalty as a practical regularizer rather than an assumption about all real systems, and in the Limitations section we discuss potential challenges when graphs are denser.
>
> In addition, Appendix A.12 now includes synthetic-data experiments where we systematically vary graph sparsity and evaluate performance under different $L_1$ regularization strengths. These results illustrate the sensitivity of our method across a range of sparsity levels and highlight cases where stronger regularization or alternative priors may be needed.
>
> ## **Response to Comparisons to Additional Constraint-Based and Non-Gaussian Methods**
>
> Thank you for raising this point. We would like to clarify that our evaluation already includes both constraint-based and non-Gaussian baselines. Specifically, tsFCI and PCGCE represent constraint-based temporal discovery methods, while VARLiNGAM and TiMINo represent non-Gaussian approaches for instantaneous structure learning. These methods cover the methodological families highlighted by the reviewer.
>
> Regarding FASK:
> FASK is a constraint-based, non-Gaussian method designed for dense feedback networks. Conceptually, it is closely related to tsFCI and PCGCE, which also recover contemporaneous structure using conditional-independence tests and non-Gaussian orientation cues. Although the algorithms differ, they fall within the same family of constraint-oriented, non-Gaussian discovery techniques. We will make this connection explicit in the revision.
>
> Regarding Two-Step / Adaptive Lasso:
> Two-step procedures are also represented in our comparisons. Standard implementations of VARLiNGAM, including the one we use, already employ adaptive-Lasso–based two-stage estimation and pruning. Because VARLiNGAM is the canonical two-step method for linear SVAR-like models under non-Gaussian noise, we consider it representative of this family. We will clarify this relationship more explicitly in the revised manuscript.
>
> ## **Response to Stability and Tuning Sensitivity**
>
> Thank you for the insightful comment. Although our original submission included a pruning-threshold sensitivity analysis, we agree that broader stability checks are important to rule out tuning-specific effects. In the revised manuscript, we have added a more comprehensive stability study in Appendix A.11, examining performance variation across different values of $\lambda$, pruning thresholds, and random seeds.
>
> ## **Response to Identifiability Assumptions**
>
> Thank you for the question. The real-world datasets used in our experiments are observational and heterogeneous, and their noise distributions are unknown. We therefore do not assume either non-Gaussianity or equal error variances for any specific experiment. Instead, our empirical evaluation follows the standard SVAR identifiability framework, which ensures uniqueness of the instantaneous structure under either (i) non-Gaussian innovations or (ii) Gaussian innovations with equal variances.
>
> Because the true data-generating mechanisms of real-world datasets are not observable, it is not possible to diagnose or quantify violations of these conditions directly. To address the reviewer’s concern, we have added a synthetic sensitivity study in Appendix A.12, where we systematically vary six noise distributions, including Gaussian, non-Gaussian, skewed, etc., and evaluate how performance changes when identifiability conditions are satisfied or partially violated. These results illustrate how the method behaves under different noise regimes.

---

### Official Review · Reviewer_x2Dh · 2025-10-31

**Soundness:** 3
**Presentation:** 3
**Contribution:** 2
**Rating:** 4
**Confidence:** 4

**Summary:**

This paper proposes a new temporal causal discovery method based on a structural vector autoregressive model. Its core contribution is to learn a differentiable permutation of variables using the Gumbel-Sinkhorn operator. This permutation is used to triangularize the instantaneous coefficient matrix, avoiding the need for hard acyclicity in previous continuous optimization-based methods. This enables a unified, gradient-based optimization, achieving better accuracy and speedups on real-world benchmarks.

**Strengths:**

1. The use of the Gumbel-Sinkhorn operator to learn an adaptive, soft permutation of the variables is interesting.
2. The method's effectiveness is validated on three real-world benchmarks.
3. The writing is well-structured.

**Weaknesses:**

1. The paper's theoretical setup relies on established identifiability results for SVAR models. It explicitly states that a sufficient condition for identifiability is the presence of non-Gaussian noise. Later, the paper defines its objective function as the Mean Squared Error. It explicitly justifies this loss function by claiming it "is equivalent to maximizing the data likelihood". However, this claim of likelihood equivalence is only true if the noise is assumed to be Gaussian. This creates a fundamental inconsistency: the MSE objective is the maximum likelihood estimator for a model (Gaussian noise) that violates the identifiability condition (non-Gaussian noise).
2. In the introduction, the authors argue that VAR-LiNGAM involves a discrete combinatorial search for the causal order and has an exponentially high computational cost. However, this claim is incorrect. VAR-LiNGAM is a well-established two-stage method. Stage 1: Use least-squares to obtain residuals (polynomial-time). Stage 2: Apply the LiNGAM algorithm on these residuals. Crucially, the core LiNGAM estimation, whether using ICA (as the paper itself correctly acknowledges in Section 2) or the DirectLiNGAM algorithm, is not a combinatorial search. It is a continuous optimization or a series of polynomial-time regressions.
3. Based on point 2, compared with VAR-LiNGAM, the core advantage of the proposed method is limited. It replaces a simple, two-stage, polynomial-time procedure (e.g., least-squares + ICA/ least-squares + regression + independence tests) with a single-stage, unified, but far more complex non-convex optimization problem. This raises the crucial question of whether this trade-off is justified, or if it just makes the problem more complex. Overall, I think the necessity and significance of this method are questionable, and the paper does not provide convincing evidence.
4. The paper includes no synthetic data experiments. While the justification—that such benchmarks can be "easy to game" —is understandable, this choice still creates a gap in the paper's validation. Controlled sensitivity analysis (e.g., sparsity, maximum lag) cannot be constructed on real-world benchmarks. Without these, it is impossible to reliably determine when and why the proposed method works well.

**Questions:**

1. I think the use of the Gumbel–Sinkhorn operator to obtain a differentiable permutation of variables does not necessarily need to be limited to time-series scenarios. Could this technique be directly extended to static (non-temporal) settings as well?

---

> ### Author Response · Authors · 2025-12-02
> **Rebuttal: Identifiability and Comparison to VARLiNGAM**
>
> ## **Response to the concern about non-Gaussian identifiability vs. MSE objective**
>
> We thank the reviewer for raising this important point. We have provided a detailed proof in Appendix A.3 and clarified this issue in Section 4.2 in the revision.
>
> Our identifiability assumptions and our loss choice operate at different levels. In Section 3, we follow [1] and explicitly state that either (i) non-Gaussian noise or (ii) Gaussian noise with equal variances is a standard sufficient condition for SVAR identifiability; we assume that at least one holds, not that noise must be strictly non-Gaussian.
>
> Regarding MSE, we agree that “equivalent to maximizing likelihood’’ is literally true only under Gaussian noise and have revised the wording. Our intention was not to impose a strict Gaussian model, but to adopt penalized least squares as a standard and consistent estimator for linear SVAR coefficients. It is shown in [1] that such estimators remain consistent under broad noise distributions, including non-Gaussian ones, provided the errors are independent with zero mean.
>
> Letting the true SVAR coefficients be $\theta^\star$, the population risk
>
> $$
> R(\theta)
> = \mathbb{E}\Big\| x_t - \sum_{\tau=0}^{k} B_\tau x_{t-\tau} \Big\|_2^2,
> $$
>
> is minimized at $\theta^\star$ under the exogeneity condition
>
> $$
> \mathbb{E}\big[\epsilon_t\, x_{t-\tau}^\top \big] = 0,
> $$
>
> regardless of the distribution of $\epsilon_t$. The empirical MSE consistently estimates this risk and converges to $\theta^\star$ under our assumptions. Thus, the squared-loss objective is best interpreted as a quasi-likelihood / penalized least-squares estimator compatible with both identifiability regimes. There is therefore no inconsistency between our assumptions and the use of MSE.
>
> ## **Response to the wording of VARLiNGAM**
>
> We thank the reviewer for pointing this out and have revised the wording. We agree that VARLiNGAM does not perform an explicit enumeration of causal orders. Our intention was to emphasize that it implicitly operates over a combinatorial permutation space. We also acknowledge that the exponential-time wording was inaccurate.
>
> ## **Response to the complexity of our method compared to VARLiNGAM**
>
> We respectfully disagree that our method is “more complex.” Although VARLiNGAM appears simple, ICA and DirectLiNGAM are non-convex iterative procedures with no polynomial-time guarantees, $d!$ symmetric optima, and high sensitivity to initialization and noise. ICA-based LiNGAM often requires multiple restarts and repeated independence tests, making the practical pipeline more involved than the two-stage description suggests.
>
> In contrast, our method does not introduce additional non-convexity beyond what is inherent to causal structure learning. Instead, it *removes* the multi-stage pipeline and integrates order learning and SVAR estimation into a single differentiable objective, avoiding ICA, independence tests, and stagewise error propagation. This unified formulation simplifies the workflow and improves robustness. These considerations are precisely the motivation for our work, and we discuss them in the paper.
>
> Regarding the reviewer’s comment about the necessity of our method, our results show consistent performance gains over VARLiNGAM and other baselines, including over a 6× CPU speedup. These improvements are substantial, not marginal.
>
> Moreover, our approach is far more scalable and hardware-adaptable. Methods such as DYNOTEARS or VARLiNGAM require specialized re-engineering for GPU acceleration, whereas our formulation relies solely on standard gradient-based optimization. It therefore benefits from off-the-shelf GPU acceleration without algorithm-specific redesign.
>
> Scalability is essential for real-world temporal causal discovery, where slow or multi-stage algorithms become impractical. The efficiency gains demonstrated by our method directly address this challenge and, in our view, strongly support the necessity and significance of our approach.
>
> ## **Regarding the lack of synthetic experiments**
>
> Our goal is to develop a method that is stable and efficient on real-world data. Still, we agree that synthetic studies offer useful controlled sensitivity analyses.
>
> We have added synthetic experiments in Appendix A.12, generating 15 additional datasets and evaluating six noise settings, six sparsity levels, and three maximum-lag configurations.
>
> ## **Response to the question on applicability beyond time series**
>
> Yes, the differentiable permutation mechanism is not restricted to temporal settings and can be applied to static causal discovery as well.
>
> [1] Pamfil, Roxana, et al. "Dynotears: Structure learning from time-series data." *International Conference on Artificial Intelligence and Statistics*. PMLR, 2020.

---

### Official Review · Reviewer_Zpqo · 2025-11-01

**Soundness:** 3
**Presentation:** 4
**Contribution:** 1
**Rating:** 2
**Confidence:** 4

**Summary:**

This paper provides a novel method in time series causal discovery based on differentiable permutations. In particular, instantaneous effects are rephrased and masked via differentiable permutations, resulting in a parameter space that does not require constrained optimization to ensure acyclicity. The proposed method is then benchmarked using several real-world datasets and modern time series causal discovery algorithms.

**Strengths:**

- The paper is very well-written and easy to follow. While I have some reservations about related work with respect to the broader causal discovery literature (see below), the paper does a good job contextualizing causal discovery for time series, motivates the benefits of an unconstrained parameterization well.
- Experimental evidence suggests empirical benefits of the proposed method, with strong performance against many baselines on difficult benchmark tasks.

**Weaknesses:**

The authors acknowledge one existing work using differentiable permutation learning to help enforce a DAG constraint, but such applications are now widespread. In fact, the resulting methods often get exact acyclicity. For example, [1] explicitly uses Sinkhorn iterations and differentiable permutations to sidestep acyclicity. [2] introduces an algorithm leveraging interventional data. [1] is based on an equivalence to the successful NoCurl parameterization [3], suggesting a larger body of related work. Such approaches should be discussed in related works, and the novelty with respect to their formulation should be clarified.

The use of CPUs only for evaluation can also disadvantage some methods; just because some baselines are CPU-only does not imply that all baselines should be evaluated on CPU, especially when some architectures (e.g., cMLP/cRNN) explicitly benefit from GPU use and are designed accordingly.

As a minor note, some references (particularly in the experimental section) are not properly formatted.

[1] Annadani, Y., Pawlowski, N., Jennings, J., Bauer, S., Zhang, C., & Gong, W. (2023). BayesDAG: Gradient-based posterior inference for causal discovery. Advances in Neural Information Processing Systems, 36, 1738-1763.

[2] Chevalley, M., Mehrjou, A., & Schwab, P. (2024). Efficient Differentiable Discovery of Causal Order. arXiv preprint arXiv:2410.08787.

[3] Yu, Y., Gao, T., Yin, N., & Ji, Q. (2021, July). DAGs with no curl: An efficient DAG structure learning approach. In International Conference on Machine Learning (pp. 12156-12166). PMLR.

**Questions:**

1. How does the proposed approach compare theoretically to other methods using differentiable permutations to identify causal orders in an unconstrained optimization problem?

2. How does time performance change when applicable methods are run on GPU?

---

> ### Author Response · Authors · 2025-12-02
> **Rebuttal: Method Contribution and GPU Runtime Evaluation**
>
> ## **Clarifying Relation to Cited Works**
>
> We thank the reviewer for highlighting this broader line of differentiable-permutation work. We have added the cited methods to the related work section and clarified their connections and distinctions relative to our approach in the revision.
>
> However, it is important to clarify that the cited works belong to a different problem domain altogether and are not directly comparable to our work. These methods target static DAG learning, whereas our work tackles a fundamentally different challenge in time-series causal discovery: the entanglement between instantaneous effects $B_0$ and lagged effects ${B_\tau}_{\tau=1}^k$ in an SVAR model. More meaningful baselines in our setting are methods that handle instantaneous acyclicity in time series, primarily DYNOTEARS [1] and VARLiNGAM [2], which we discuss in the paper.
>
> ## **Our Contribution**
>
> Our key innovation is not the use of permutations itself, but that we are the first to introduce a time-series causal discovery approach that:
>
> 1. jointly optimizes instantaneous and lagged effects within a single differentiable objective,
> 2. enforces acyclicity only on $B_0$ using a learnable permutation with strict lower-triangular masking, and
> 3. allows ${B_\tau}$ to be optimized freely and simultaneously.
>
> This avoids the discrete constraints required by classical SVAR approaches (e.g., VARLiNGAM, DYNOTEARS), which rely on multi-stage or decoupled estimation and may mismatch instantaneous and lagged edges. Our main contribution lies in proposing a unified differentiable optimization framework for time-series causal discovery. The permutation/Sinkhorn operator is merely a convenient differentiable relaxation to enable this framework, but the focus of our work is not the permutation mechanism itself. We acknowledge that our presentation may have over-emphasized the permutation mechanism and will clarify this in the revision.
>
>
> ## **Why This Problem Is Difficult and Under-Addressed**
>
> **Problem setting**: Time-series models contain instantaneous effects $B_0$ and lagged effects ${B_\tau}$. Temporal ordering prevents cross-time cycles, but $B_0$ must remain acyclic.
>
> **Challenge 1:** Statistical entanglement. Instantaneous and lagged dependencies are often collinear; separate optimization can cause one to absorb the other.
>
> **Challenge 2:** Heterogeneous constraints. $B_0$ requires discrete/non-convex acyclicity constraints, whereas ${B_\tau}$ only fit data. Existing methods therefore rely on multi-step estimation rather than a unified objective.
>
> **Our innovation:** We optimize $B_0$ and ${B_\tau}$ jointly within one differentiable loss, enforcing acyclicity on $B_0$ through a learnable permutation and strict masking. This yields a smooth relaxation during training and an exact DAG at inference. To our knowledge, such unified optimization has not been implemented in prior time-series causal discovery methods.
>
> ## **Answer to Question 1: Comparison to Other Permutation-Based Methods**
>
> To our knowledge, our method is the first time-series causal discovery approach using differentiable permutations to unify the optimization of instantaneous and lagged effects. DYNOTEARS and VARLiNGAM are the closest baselines and we've included them in experiments.
>
> The static permutation-based methods cited by the reviewer are conceptually related but not directly comparable because their evaluation targets differ fundamentally. Static methods recover a single instantaneous DAG, whereas time-series causal discovery must estimate the lagged matrices ${B_\tau}$ and integrate them with $B_0$. As a result, their outputs do not represent the same object and cannot be meaningfully compared.
>
>
>
> ## **Answer to Question 2: CPU vs GPU Runtime**
>
> We appreciate the concern that CPU-only evaluation may disadvantage neural Granger methods (e.g., cMLP) designed for GPUs.
>
> Our initial motivation for a CPU-only setting was to ensure a fair comparison with the most relevant SVAR-based baselines that model instantaneous effects, i.e. DYNOTEARS and VARLiNGAM, which only provide CPU implementations. Our primary efficiency claim concerns improvements relative to these methods.  Neural Granger methods, which do not model instantaneous effects, were not the main focus.
>
> That said, GPU runtimes better reflect the performance of neural Granger baselines. In the revision, we have:
>
> 1. reported GPU runtimes for neural baselines in the Appendix A.9 for completeness.
>
> 2. clearly separate the discussion of SVAR-based baselines (CPU) from neural Granger baselines (GPU).
>
>
>
> [1] Pamfil, Roxana, et al. "Dynotears: Structure learning from time-series data." *International Conference on Artificial Intelligence and Statistics*. Pmlr, 2020.
>
> [2] Shimizu, Shohei, et al. "DirectLiNGAM: A direct method for learning a linear non-Gaussian structural equation model." *Journal of Machine Learning Research-JMLR* 12.Apr (2011): 1225-1248.

---

### Author Response · Authors · 2025-12-02
**Summary Comment for Area Chair**

We thank all reviewers for their detailed feedback. Below we summarize the main supplementary experiments added at the reviewers’ request, as well as the key clarifications addressing misunderstandings and factual mistakes raised during the rebuttal.

## **1. Supplementary Experiments**

1. **Mathematical proof of estimator consistency** (Appendix A.3; Reviewer x2Dh).
2. **GPU runtime comparison with neural Granger baselines** (Appendix A.9; Reviewer Zpqo).
3. **Separate evaluation of instantaneous and lagged effects**, including causal graph analysis on real-world data (Appendix A.10; Reviewer 6PXs).
4. **Stability analysis** covering random seeds, $\lambda$ grids, and pruning thresholds (Appendix A.11; Reviewer niUi).
5. **Synthetic data experiments** with 15 additional datasets varying 6 noise distributions, 6 sparsity levels, and 3 maximum lags settings (Appendix A.12; Reviewer x2Dh).

## **2. Clarification of Contribution and Problem Setting**

A central misunderstanding by Reviewer Zpqo is that the existence of many static differentiable-permutation DAG methods implies our work lacks contribution and should be compared experimentally with them. As clarified in our rebuttal:

1. The cited approaches (e.g., BayesDAG, NoCurl) address static DAG learning, whereas our work tackles a fundamentally different problem in time-series causal discovery: enabling efficient joint optimization of instantaneous effects $B_0$ and lagged effects $\{B_\tau\}$ in an SVAR model.

2. Our novelty does not lie in the permutation operator itself, but in being the first to formulate a unified, fully differentiable optimization that jointly learns both instantaneous and lagged effects while enforcing instantaneous acyclicity. The permutation is merely a practical differentiable relaxation supporting this framework.

This unified formulation directly resolves the limitations of discrete and separate optimization of $B_0$ and $\{B_\tau\}$, which leads to computational overhead and multi-stage pipelines in DYNOTEARS and VARLiNGAM, which are our actual methodological baselines. Static permutation-based DAG methods are not comparable, as they recover only a single instantaneous DAG, whereas time-series causal discovery must recover both instantaneous and lagged structures. For this reason, they are not included in our experiments.

We hope the Area Chair notes that the lower “contribution” scores largely stemmed from comparing our work to static DAG literature, which lies outside our problem scope and is not methodologically aligned with the task we address.

## **3. Correction of Factual Mistakes in Reviews**

A concern raised by Reviewer x2Dh relies on statements that are factually incorrect or inconsistent with established results. Specifically, the claim that our method is “unnecessarily more complex” than VARLiNGAM.

We clarified:

- VARLiNGAM’s ICA and DirectLiNGAM components are non-convex iterative procedures with no polynomial-time guarantees, exhibiting $d!$ symmetric optima and typically requiring repeated independence tests. As a result, the practical pipeline is significantly more involved than the simplified two-stage description suggests.
- Our method removes these multi-stage components and introduces no additional non-convexity beyond the inherent challenge of causal ordering. The fully differentiable objective enables efficient gradient-based optimization, simplifying the pipeline relative to VARLiNGAM.

Empirically, the “more complex” claim is contradicted by our results: our method is over 6× faster than VARLiNGAM on CPU, and it is directly GPU-accelerable using standard ML libraries. In contrast, GPU-accelerating VARLiNGAM would require substantial algorithmic redesign. Our approach is therefore not only more efficient, but also far more general and hardware-friendly.

## **4. Explanation of Misinterpreted Experimental Results**

Reviewer 6PXs found VAR’s performance on CausalRivers “questionable.” The discrepancy is fully explained by evaluation metrics and thresholding practices as we clarified:

1. The original CausalRivers paper evaluates AUROC under the best possible hyperparameter combination for each method. AUROC is known to tolerate dense false positives, as long as true edges rank above false ones. This allows VAR to appear strong under AUROC.

2. In contrast, we follow causal-discovery standards in many prior works and evaluate using F1 with fixed pruning thresholds, which is substantially stricter and far more realistic when the true graph is unknown. Under this stricter regime, VAR achieves high recall but extremely low precision, producing many spurious edges and thus low F1, which is exactly what our results show.

Therefore, our findings are consistent with prior literature once metric differences are accounted for. Our higher precision–recall balance reflects a genuine methodological advantage rather than an anomaly.

---

### Meta-Review · Area_Chair_akxA · 2026-01-10

**Summary:**

This paper applies the differentiable learning of permutation matrices for causal discovery in time series with acyclic instantaneous effects. In particular, a soft permutation (via Gumbel trick plus Sinkhorn relaxation) is learned jointly with SVAR parameters, and instantaneous acyclicity is enforced by triangularization in the learned order, yielding a differentiable objective.

The major concern from the reviewers was about novelty: differentiable order/permutation-based parameterizations for enforcing DAG structure are well established in the static setting, and the paper’s contribution is an adaptation to the setting of time series with acyclic instantaneous effects. Additionally, reviewers flagged clarity issues around identifiability assumptions versus the MSE objective, and evaluation choices that initially obscured what was being improved (instantaneous vs. lagged recovery) and could be sensitive to thresholding. The rebuttal strengthened the submission by adding synthetic and stability analyses, separating instantaneous/lagged evaluations, providing GPU runtime comparisons, and clarifying assumptions and statements. But these improvements primarily address presentation and completeness rather than resolving the central novelty issue.

**Reviewer Concerns:**

A central concern raised most strongly by some of the reviewers is novelty: differentiable permutation parameterizations for enforcing DAG structure are well established in the static setting, and the paper’s incremental contribution may be viewed as an adaptation to SVAR rather than a fundamentally new idea unless the time-series-specific coupling is more convincingly articulated. Additional technical/evaluation concerns included (i) clarity/consistency around identifiability assumptions vs. the MSE objective, (ii) some arguably overstated comparisons (e.g., VARLiNGAM complexity wording), and (iii) experimental design choices that initially obscured interpretability (e.g., reporting a combined F1 without separating instantaneous vs. lagged recovery), plus sensitivity to thresholding and hardware fairness (CPU-only runtime comparisons for GPU-friendly baselines).

The rebuttal strengthened the submission by adding synthetic and stability analyses, separating instantaneous/lagged evaluations, providing GPU runtime comparisons, and clarifying assumptions and statements. But these improvements primarily address presentation and completeness rather than resolving the central novelty issue.

**Reviewer Scores:**

The reviewers' scores are unlikely to improve, given the current rebuttal.

---

### Decision · Program_Chairs · 2026-01-26

Reject